

# The contribution of residential coal combustion to atmospheric PM$_{2.5}$

# in the North China during winter

**Pengfei Liu**[1, 3], **Chenglong Zhang**[1, 3], **Chaoyang Xue**[1, 3], **Yujing Mu**[1, 2, 3], **Junfeng Liu**[1, 3], **Yuanyuan Zhang**[1, 3], **Di Tian**[1, 3], **Can Ye**[1, 3], **Hongxing Zhang**[1, 4], **Jian Guan**[5]

[1] Research Center for Eco-Environmental Sciences, Chinese Academy of Sciences, Beijing, 100085, China

[2] Center for Excellence in Urban Atmospheric Environment, Institute of Urban Environment, Chinese Academy of Sciences, Xiamen, 361021, China

[3] University of Chinese Academy of Sciences, Beijing, 100049, China

[4] Beijing Urban Ecosystem Research Station, Beijing, 100085, China

[5] Environment Monitoring Station of Baoding City, Hebei, 071000, China

*Correspondence to:* **Y. J. Mu** (yjmu@rcees.ac.cn)

Abstract: The vast area in the North China, especially during wintertime, is currently suffering from severe haze events due to the high levels of atmospheric PM$_{2.5}$. To recognize the reasons for the high levels of PM$_{2.5}$, daily samples of PM$_{2.5}$ were simultaneously collected at the four sampling sites of Beijing City (BJ), Baoding City (BD), Wangdu County (WD) and Dongbaituo Countryside (DBT) during the winters and springs of 2014-2015. The concentrations of the typical water-soluble ions (WSIs, such as Cl$^-$, NO$_3^-$, SO$_4^{2-}$ and NH$_4^+$) at DBT were found to be remarkably higher than those at BJ in the two winters but almost the same as those at BJ in the two springs. The evidently greater concentrations of OC, EC and secondary inorganic ions (NO$_3^-$, SO$_4^{2-}$, NH$_4^+$ and Cl$^-$) at DBT than at WD, BD and BJ during the winter of 2015 indicated that the pollutants in the rural area were not due to transportation from its neighbor cities but dominated by local emissions. As the distinct source for atmospheric OC and EC in the rural area, the residential coal combustion also made contribution to secondary inorganic ions through the emissions of their precursors (NO$_x$, SO$_2$, NH$_3$ and HCl) as well as heterogeneous or multiphase reactions on the surface of OC and EC. The average mass proportions of OC, EC, NO$_3^-$ and SO$_4^{2-}$ at BD and WD were found to be very close to those at DBT, but evidently different from those at BJ, implying that the pollutants in the cities of WD and BD which are fully surrounded by the countryside were strongly affected by the residential coal combustion. The OC/EC ratios at the four sampling sites became the almost same value of 4.8 when the concentrations of PM$_{2.5}$ were greater than 150 μg m$^{-3}$, suggesting that the residential coal combustion could also make dominant contribution to atmospheric PM$_{2.5}$ at BJ during the severe pollution period when the air parcels were usually from southwest–south regions where high density of farmers reside. The evident increase of the number of the species involved in significant correlations from the countryside to the cities further confirmed that residential coal combustion was preferentially dominant source for the key species in the rural area whereas the complex sources including local emissions and regional transportation were dominant for atmospheric species in the cities. The significant correlations among OC, EC, Cl$^-$, NO$_3^-$, and NH$_4^+$ were found at the four sampling sites but only significant correlation between OC (or EC) and SO$_4^{2-}$ was found at BJ, implying that the formation rate of SO$_4^{2-}$ via heterogeneous or multiphase reactions might be relatively slower than those of NO$_3^-$, NH$_4^+$ and Cl$^-$. Based on the chemical mass closure (CMC) method, the contributions of the





primary particle emission from residential coal combustion to atmospheric PM$_{2.5}$ at BJ, BD, WD
and DBT were estimated to be 32 %, 49 %, 43 % and 58 %, respectively.

**1 Introduction**

In recent years, the vast area in the North China is frequently suffering from severe haze pollution
(Chan and Yao, 2008; Zhang et al., 2012; Zhang et al., 2015), which has aroused great attention to
the public (Guo et al., 2014; Huang et al., 2014; Cheng et al., 2016; Wang et al., 2016; J. Liu et al.,
2016). The severe haze pollution is mainly due to the high level of fine particulate matters with
dynamic diameter less than 2.5μm (PM$_{2.5}$) (Huang et al., 2014; P. Liu et al., 2016). PM$_{2.5}$ can
reduce atmospheric visibility by absorbing or scattering the incident light (Buseck and Posfai,
1999; Cheng et al., 2006) and increase morbidity and mortality by penetrating the human bronchi
and lungs (Nel, 2005; Poschl, 2005; Peplow, 2014).
To alleviate the serious haze pollution problems, the Chinese government has performed a series
of control measures for major pollution sources (Zhang et al., 2012; J. Liu et al., 2016; Li et al.,
2016b; Wen et al., 2016). For example, coal-fired power plants have been forced to install flue gas
desulfurization and denitration (Zhang et al., 2012; Chen et al., 2014), coal has been replaced with
natural gas and electricity in megacities (Wang et al., 2009; Duan et al., 2012; Zhao et al., 2013a;
Tan et al., 2016), stricter emission standards have been implemented for vehicles and industrial
boilers (Zhang et al., 2012; Tang et al., 2016) and so on, resulting in the decrease trend of primary
pollutants including PM$_{2.5}$ in recent years (Ma et al., 2016; Wen et al., 2016; Zhang et al., 2016).
However, the PM$_{2.5}$ levels still achieved to be above 1000 μg m$^{-3}$ in some areas of
Beijing-Tianjin-Hebei (BTH) region during the period of the red alert for haze in December 2016
(http://english.mep.gov.cn/News_service/media_news/201612/t20161220_369317.shtml)   when
the stricter control measures (e.g. stop production for industries and construction, and the odd and





even number rule) had been performed (Y. Li et al., 2016), implying that sources other than
industries, construction and vehicles might make dominant contribution to atmospheric $PM_{2.5}$ in
the region. Residential coal combustion which is prevailing for heating during winter in the region
was suspected to be a dominant source for atmospheric $PM_{2.5}$. Although annual residential coal
consumption (about 4,200,000 kg) in BTH region only accounts for small fraction (about 11 %) of
the total coal consumption (http://www.qstheory.cn/st/dfst/201306/t20130607_238302.htm), the
emission factors of primary pollutants including $PM_{2.5}$ from the residential coal combustion have
been found to be about 1-3 orders of magnitude greater than those from coal combustion of
industries and power plants (Revuelta et al., 1999; Chen et al., 2005; Xu et al., 2006; Zhang et al.,
2008; Geng et al., 2014; Yang et al., 2016). In addition, annual residential coal consumption
mainly focuses on the four months in winter. Although the Chinese government has implemented
control measures for residential coal combustion (e.g. replacement of traditional coal stoves by
new stoves, bituminous coal by anthracite, and coal by electricity and natural gas), the promotion
strength of the control measures was still very limited. Additionally, the promotion new stoves are
still with strong smoke emission due to lack of clean combustion technique, and the anthracite is
not welcomed by farmers because of its extremely slow combustion rate in comparison with
bituminous coal.
There were few studies focusing on the influence of residential coal combustion on atmospheric
particles in the North China. W. Li et al. (2014) concluded that strong sources for $PM_{10}$ in rural
residential areas were from household solid fuel combustion, based on annual mean $PM_{10}$
concentrations observed in urban regions ($180 \pm 171$ μg m$^{-3}$) and rural villages ($182 \pm 154$ μg m$^{-3}$)
in the northern China. Duan et al. (2012) inferred that the lower OC/EC ratios at the rural site than



at the urban site was ascribed to coal combustion prevailed in the rural area. Our previous study
revealed that residential coal combustion made evident contribution to atmospheric water-soluble
ions (WSIs) in Beijing (P. Liu et al., 2016). Based on Weather Research and Forecasting model
coupled with Chemistry, J. Liu et al. (2016) recently estimated that the residential sources (solid
fuel) contributed 32 % and 53 % of the primary $PM_{2.5}$ emissions in the BTH region during the
whole year and during the winter of 2010, respectively.
In this study, daily samples of $PM_{2.5}$ were simultaneously collected at the four sampling sites
(Beijing City, Baoding City, Wangdu County and Dongbaituo Countryside) during the winters and
springs of 2014-2015, and the direct evidence for the influence of residential coal combustion on
regional $PM_{2.5}$ in the region was found based on the $PM_{2.5}$ levels, the $PM_{2.5}$ composition
characteristics, the correlations among the key species in $PM_{2.5}$, the back trajectories and the
chemical mass closure method.
**2 Materials and methods**
**2.1 Sampling sites**
The two sampling sites in Beijing City and Dongbaituo Countryside, which have been described
in detail by our previous study (P. Liu et al., 2016), were selected on a rooftop (approximately 25
m and 5 m above ground, respectively) of the Research Center for Eco-Environment Sciences,
Chinese Academy of Sciences (RCEES, CAS) and a field station in the agricultural field of
Dongbaituo village, Baoding, Hebei Province, respectively. Another two sampling sites in
Baoding City and Wangdu County were both chosen on the rooftop of local environmental
monitor station (about 30 m and 20 m above ground, respectively), which are both located in the
center of the cities and surrounded by some commercial and residential areas. The detailed



location of the four sampling sites is presented in Fig. 1 and the distances between Beijing and
Baoding, Baoding and Wangdu, Wangdu and Dongbaituo are about 156 km, 36 km and 12 km,
respectively. Thereafter, the sampling sites of Beijing, Baoding, Wangdu and Dongbaituo are
abbreviated as BJ, BD, WD and DBT, respectively.
**2.2 Sample collection and analysis**
$PM_{2.5}$ samples at BJ and DBT were collected simultaneously on PTFE filters (90 mm, Millipore)
by medium-volume $PM_{2.5}$ samplers (LaoYing-2034) at a flow rate of 100 L $min^{-1}$ from January 15,
2014 to May 31, 2015, in winter (January 15, 2014-Febrary 25, 2014, November 18, 2014-January
20, 2015 and February 11, 2015-March 15, 2015) and spring (April 21, 2014-May 4, 2014 and
March 20, 2015-May 31, 2015). An enhanced observation which added the other two sampling
sites of BD and WD was carried and $PM_{2.5}$ samples at the four sampling sites were collected in the
same way on the quartz fiber filters (90 mm, Munktell) from January 21 to February 10, 2015. The
sampling duration was 24 h (from 15:00 p.m. to 15:00 p.m. of the following day in local time
(UTC + 8)). All the samples were put in appropriative dishes (90 mm, Millipore) after sampling
and preserved in a refrigerator immediately until analysis.
As for the quartz fiber filters, half of each filter was extracted ultrasonically with 10 mL ultrapure
water for half an hour. The solutions were filtered through a micro-porous membrane (pore size,
0.45 μm; diameter, 13 mm) before analysis and the WSIs ($Cl^-$, $NO_3^-$, $SO_4^{2-}$, $Na^+$, $NH_4^+$, $Mg^{2+}$, $Ca^{2+}$
and $K^+$) in the treated filtrates were analyzed by Ion Chromatography (IC, WAYEE IC6200) which
has been described in detail by our previous study (P. Liu et al., 2016). A quarter of each filter was
cut into fragments and digested with 5 mL 65 % $HNO_3$ and 2 mL 30 % $H_2O_2$ (Li et al., 2015) by a
microwave digestion system (SINEO, MASTER-40). The digestion solution was diluted to 25 mL





with ultrapure water to insure the solution acidity below 10 % and the trace elements (Al, Mn, Fe,
Cu, Zn, As, Se, Sr, Tl and Pb) in the diluted solution were analyzed by a triple-quadrupole
inductively coupled plasma mass spectrometry (ICP-MS/MS, Agilent 8800). The standard
reference material (GBW07427) was also digested in the same way as the samples and the
recoveries of the trace elements were within the allowable ranges of the certified values (100 ±
15 %). Another quarter of each filter was analyzed by a DRI thermal optical carbon analyzer
(DRI-2001A) for carbon components (OC and EC). In addition, the PTFE filters were only used
for analyzing the WSIs (P. Liu et al., 2016).
**2.3 Chemical mass closure**
Chemical mass closure (CMC) method was adopted by considering secondary inorganic aerosols
(SIA, the sum of $SO_4^{2-}$, $NO_3^-$ and $NH_4^+$), sea salt & coal combustion (derived from $Cl^-$ and $Na^+$),
biomass burning (characterized by $K^+$), mineral dust, EC, primary organic carbon (POC),
secondary organic carbon (SOC) and trace element oxide (TEO) (Hsu et al., 2010b; Zhang et al.,
2013; Mantas et al., 2014; Tian et al., 2014; Kong et al., 2015).
Atmospheric $Na^+$ and $Cl^-$ were considered to be from both sea salt and coal combustion during
winter in the North China (Brewer, 1975; van Eyk et al., 2011; Bläsing and Müller, 2012; Yu et al.,
2013; Wu et al., 2014; He et al., 2015; P. Liu et al., 2016), and their mass concentrations followed
the four equations:
$$[Cl^-_{cc}] + [Cl^-_{ss}] = [Cl^-] \tag{1}$$
$$[Na^+_{cc}] + [Na^+_{ss}] = [Na^+] \tag{2}$$
$$\frac{[Cl^-_{cc}]/35.5}{[Na^+_{cc}]/23} = 1.4 \tag{3}$$





$$\frac{[Cl^-_{ss}]/35.5}{[Na^+_{ss}]/23} = 1.18 \qquad (4)$$

where $[Cl^-_{ss}]$ and $[Na^+_{ss}]$ are the mass concentrations of $Cl^-$ and $Na^+$ from sea salt, and $[Cl^-_{cc}]$ and
$[Na^+_{cc}]$ are the mass concentrations of $Cl^-$ and $Na^+$ from coal combustion. The molar ratio of $Cl^-_{ss}$
to $Na^+_{ss}$ was adopted to be 1.18 which represented the typical ratio from sea salt (Brewer, 1975).
The molar ratio of $Cl^-_{cc}$ to $Na^+_{cc}$ was chosen to be 1.4 in this study according to our preliminary
measurements from the raw bituminous coal prevailed in the North China and the value of 1.4 has
been recorded by the previous study (Bläsing and Müller, 2012). If the molar ratios of atmospheric
$Cl^-$ to $Na^+$ in $PM_{2.5}$ were greater than the value of 1.4 or lower than the value of 1.18, atmospheric
$Cl^-$ and $Na^+$ would be considered to be totally from coal combustion or sea salt.
Because the average Al content accounts for about 7 % in mineral dust (Zhang et al., 2003; Ho et
al., 2006; Hsu et al., 2010a; Zhang et al., 2013), the mineral dust was estimated based on the
follow equation:
$$[Mineral\ dust] = \frac{[Al]}{0.07} \qquad (5)$$

POC and SOC were calculated by the EC-tracer OC/EC method (Cheng et al., 2011; Zhao et al.,
2013b; G. J. Zheng et al., 2015; Cui et al., 2015) as follows:
$$[POC] = [EC] \times ([OC]/[EC])_{pri} = K[EC] + M \qquad (6)$$

$$[SOC] = [OC] - [POC] \qquad (7)$$

The values of $K$ and $M$ are estimated by linear regression analysis using the data pairs with the
lowest 10 % percentile of ambient OC/EC ratios.
To estimate the contribution of heavy metal oxide, the enrichment factors (EF) of various heavy
metal elements were calculated by the following equation (Hsu et al., 2010b; Zhang et al., 2013):



$$EF = \frac{([Element]/[Al])_{aerosol}}{([Element]/[Al])_{crust}}$$ (8)
where $([Element]/[Al])_{aerosol}$ is the ratio of the element to Al in aerosols and $([Element]/[Al])_{crust}$ is
the ratio of the element to Al in the average crust (Taylor, 1964). According to the method
developed by Landis et al. (2001), the atmospheric concentrations of elements were multiplied by
a factor of 0, 0.5 and 1 if their EFs were less than 1, between 1 and 5, and greater than 5,
respectively. Based on the EFs (Fig. 2), the equation for estimating TEO was derived as following:
$$[TEO] = 1.3 \times ([Cu] + [Zn] + [Pb] + [As] + [Se] + [Tl] + 0.5 \times [Mn])$$ (9)
The value of 1.3 was the conversion factor of metal abundance to oxide abundance. It should be
mentioned that some other elements such as Cd and Ba were not measured in this study, probably
resulting in underestimating the proportion of TEO. Nevertheless, the biases are probably
insignificant because the proportion of TEO only accounted for less than 2 % in $PM_{2.5}$.
**2.4 Meteorological, trace gases and back trajectory**
Both the meteorological data, including wind speed, wind direction, relative humidity (RH),
temperature, barometric pressure and air quality index (AQI) of $PM_{2.5}$, $SO_2$, $NO_2$, CO, $O_3$ at BJ,
BD and WD were obtained from Beijing urban ecosystem research station in RCEES, CAS
(http://www.bjurban.rcees.cas.cn/), environmental protection bureau of Baoding City
(http://bdhb.gov.cn/) and environmental monitoring station of Wangdu County
(http://www.wdx.gov.cn/), respectively. The meteorological data at BJ and BD is shown in Fig. 3
and the average concentrations of $SO_2$ and $NO_2$ at BJ, BD and WD are listed in Table 2 during the
sampling period in the winter of 2015, which would be discussed in section 3.2 and 3.3.
The air mass backward trajectories were calculated for 24 h through the National Oceanic and
Atmospheric Administration (NOAA) Hybrid Single-Particle Lagrangian Integrated Trajectory





Version 4 model (HYSPLIT 4 model) with National Centers for Environmental Prediction's
(NCEP) global data. The backward trajectories arriving at 500 m above sampling position were
computed at 0:00 h, 6:00 h, 12:00 h and 18:00 h (UTC) for each sampling day, respectively. A
K-means cluster method was then used for classifying the trajectories into several different
clusters and suitable clusters were selected for further analysis.
**3 Results and discussion**
**3.1 Comparison of atmospheric WSIs between BJ and DBT**
The daily variations of atmospheric WSIs during the sampling periods at the two sampling sites of
BJ and DBT are shown in Fig. 4. It is evident that the variations of the WSIs between the two
sampling sites of BJ and DBT exhibited similar trend, but the mass concentrations of the WSIs
were remarkably greater at DBT than at BJ during the two winter seasons. As listed in Table 1, the
average concentrations of the typical WSIs were a factor of 1.5-2.0 greater at DBT than at BJ
during the two winter seasons, whereas they were approximately the same at the two sampling
sites during the two spring seasons. To clearly reveal the differences, the daily D-values (the
concentrations of WSIs at DBT minus those at BJ) of several typical WSIs as well as the total
WSIs between DBT and BJ are individually illustrated in Fig. 5. With only the exception for $Ca^{2+}$
(typical mineral dust component), the D-values of $NH_4^+$, $NO_3^-$, $SO_4^{2-}$ and $Cl^-$ between DBT and BJ
exhibited obviously positive values during the most sampling days in the two winter seasons,
implying that the sources related to mineral dust could be excluded for explaining the obviously
higher concentrations of the WSIs at DBT than at BJ. The sampling site of DBT is adjacent to
Baoding city where the AQI during the winter always ranked the top three among Chinese cities in
recent years (http://113.108.142.147:20035/emcpublish/ ), and hence the relatively greater





concentrations of the WSIs at DBT might be due to the regional pollution. However, the emissions
of pollutants from industries, power plants and vehicles are usually relatively stable, which could
not account for the remarkable differences of the D-values between the winters and the springs
(Fig. 5). If the relatively high concentrations of the WSIs at DBT during the winter were ascribed
to the regional pollution, there would be additional strong sources for them in the area of Baoding.
To explore whether the regional pollution was responsible for the relatively high concentrations of
the WSIs at DBT in winter, the various species in $PM_{2.5}$ collected simultaneously at DBT and its
neighbor cities of WD, BD and BJ in the winter of 2015 were further investigated in the following
section.
**3.2 Daily variations of the species in $PM_{2.5}$ at the four sampling sites**
The daily variations of the species in $PM_{2.5}$ at the four sampling sites also exhibited similar
fluctuation trends (Fig. 6), implying that the regional meteorological conditions which are
dominant factors for the dispersion and accumulation of atmospheric pollutants (Xu et al., 2011;
Tao et al., 2012; Sun et al., 2013; Chen et al., 2015; Gao et al., 2016) were similar (Fig. 3) during
the sampling period. However, there was obvious difference in the concentrations of OC, EC,
$NH_4^+$, $NO_3^-$, $SO_4^{2-}$, $Cl^-$ and $K^+$ among the four sampling sites, ranked in order as BJ < WD < BD <
DBT. As listed in Table 2, the average concentration of the total species at DBT was about a factor
of 2.7, 1.8 and 1.4 higher than those at BJ, WD and BD, respectively. The largest levels of the key
species in $PM_{2.5}$ at DBT among the four sampling sites implied that the pollutants at the rural site
were not through the air parcel transportation from its neighbor cities but mainly ascribed to the
local emissions or formation. Vehicles and industries could be rationally excluded for explaining
the largest levels of the key species in $PM_{2.5}$ at DBT, because these sources are very sparse in the





rural area around DBT (See section 3.4). Compared with the cities, the distinct source for
atmospheric pollutants at DBT in winter is the residential coal combustion which is prevailingly
used for heating and cooking in rural areas of the Northern China. The emissions of various
pollutants from residential coal combustion were very serious due to lack of any control measures,
strong smoke could be seen in the chimney of the residential coal stoves. The emission factors of
OC and EC from residential coal combustion were reported to be 0.47-7.82 g kg$^{-1}$ coal and
0.028-2.75 g kg$^{-1}$ coal, respectively (Chen et al., 2005; Zhang et al., 2008). The emission factors of
various pollutants from a typical residential coal stove fueled with raw bituminous coal were also
investigated in our group (Du et al., 2016; Liu et al., 2017) according to farmers' customary uses
of coal stoves under the alternation cycles of flaming and smoldering. The emission factors of OC
and EC under the entire combustion process could achieve to be 10.99 $\pm$ 0.95 g kg$^{-1}$ coal and 0.84
$\pm$ 0.06 g kg$^{-1}$ coal, respectively (Table 3). Considering the high density of farmers in the rural area,
the largest levels of atmospheric OC and EC at DBT could be rationally ascribed to residential
coal combustion. However, the proportion of the WSIs from residential coal combustion (Fig. 7a)
were extremely low with respect to that of the atmosphere. Therefore, the largest levels of the key
WSIs in PM$_{2.5}$ at DBT were suspected to the secondary formation via the heterogeneous or
multiphase reactions which might be accelerated by the OC and EC (Han et al., 2013; Zhao et al.,
2016) emitted from residential coal combustion.
Although the three sampling sites of DBT, WD and BD are closely adjacent, the lowest
concentrations of the key species in PM$_{2.5}$ were observed at WD, which was probably ascribed to
the replacement of coal with natural gas for the central heating in the county of WD (a main pipe
of natural gas is just across the county), e.g., the average concentration of NO$_2$ was higher at WD



than at BD, whereas the average concentration of $SO_2$ was on the contrary (Table 2).
The city of BD and the county of WD are fully surrounded by high density of countryside,
whereas the city of BJ is only neighbored with high density of countryside in the
south-southeast-southwest directions, and thus the residential coal combustion was also suspected
to be responsible for the remarkably higher concentrations of the key species in $PM_{2.5}$ at BD and
WD than at BJ. To confirm the above assumptions, the chemical composition and source
characteristics of the species in $PM_{2.5}$ were further analyzed in the following section.
**3.3 Chemical composition of $PM_{2.5}$ at the four sampling sites**
The average mass proportions of the species in $PM_{2.5}$ during the sampling period at the four
sampling sites are illustrated in Fig. 7b. OC, EC, $NH_4^+$, $NO_3^-$ and $SO_4^{2-}$ were found to be the
principal species, accounting for about 82 %-88 % of the total species in $PM_{2.5}$ at each sampling
site, which were in line with previous studies (Zhao et al., 2013a; X. J. Zhao et al., 2013; Tian et
al., 2014; Huang et al., 2014). As for the proportions of individual species, there were obvious
differences between the sampling site of BJ and the sampling sites of BD, WD and DBT. The
average mass proportions of OC and EC at BD, WD and DBT were very close, accounting for
about 45.7 %-47.1 % and 9.0 %-10.4 % of the total species in $PM_{2.5}$, respectively, which were
about 8 % for OC and 2 % for EC greater than those at BJ. In contrast to OC and EC, the average
mass proportions of $NO_3^-$ (10.1 %-10.8 %) and $SO_4^{2-}$ (11.2 %-11.7 %) at BD, WD and DBT were
about 5 % and 3 % less than those (15.1 % for $NO_3^-$ and 14.0 % for $SO_4^{2-}$) at BJ, respectively. The
obvious differences of the mass proportions of OC, EC, $NO_3^-$ and $SO_4^{2-}$ between the sampling site
of BJ and the sampling sites of BD, WD and DBT indicated that the sources for the principal
species at BJ were different from the other three sampling sites. The mass proportions of OC, EC,



$NO_3^-$ and $SO_4^{2-}$ at BD and WD were very close to those at DBT, implying that residential coal
combustion might also be the dominant source for the species in $PM_{2.5}$ at BD and WD. Residential
sector (dominated by residential coal combustion) in the region of BTH during winter has been
recognized as the dominant source for atmospheric OC and EC, which was estimated to contribute
85% and 65% of primary OC and EC emissions, respectively (J. Liu et al., 2016). Because the
sampling sites of DBT, BD and WD are located in or fully surrounded by high density of
countryside, the contribution of residential coal combustion to atmospheric OC and EC at DBT,
BD and WD must evidently exceed the regional values estimated by J. Liu et al. (2016).
Although the mass proportions of $NO_3^-$ and $SO_4^{2-}$ were evidently lower at BD, WD and DBT than
at BJ, the average mass concentrations of $NO_3^-$ and $SO_4^{2-}$ were on the contrary (Table 2).
Atmospheric $NO_3^-$ and $SO_4^{2-}$ are mainly from secondary formation via heterogeneous, multiphase
or gas-phase reactions which are depended on the concentrations of their precursors ($NO_2$ and $SO_2$)
and OH radicals, the surface characteristics and areas of particles, and RH (Ravishankara, 1997;
Wang et al., 2013; Quan et al., 2014; Nie et al., 2014; He et al., 2014; Yang et al., 2015; B. Zheng
et al., 2015). The remarkably higher concentrations of $NO_2$, $SO_2$ and $PM_{2.5}$ at BD, WD and DBT
(Liu et al., 2015) than at BJ (Table 2) favored secondary formation of $NO_3^-$ and $SO_4^{2-}$, resulting in
the relatively high concentrations of $NO_3^-$ and $SO_4^{2-}$.
As shown in Fig. 8, the serious pollution episodes at BJ usually occurred during the periods with
the air parcel from the southwest-south directions where farmers with high density reside, and thus
residential coal combustion might also make evident contribution to atmospheric pollutants at BJ.
Because the species in $PM_{2.5}$ at BJ during the serious pollution episodes accounted for very large
weight of their average concentrations, the proportions of the species in $PM_{2.5}$ were dominated by



the serious pollution events. The highest $NO_3^-$ and $SO_4^{2-}$ proportions and the lowest OC and EC
proportions at BJ among the four sampling sites might be partly ascribed to the conversions of
$NO_2$ and $SO_2$ to $NO_3^-$ and $SO_4^{2-}$ during the air parcel transportation from the south-southwest
directions. The contribution of the transportation to atmospheric OC and EC at BJ could be
verified by the correlations between the OC/EC ratios and the $PM_{2.5}$ levels (Fig. 9). The OC/EC
ratios (about 4.9 ± 0.7) at WD and DBT were almost independent of the $PM_{2.5}$ levels, whereas the
OC/EC ratios at BJ and BD remarkably decreased with increasing the $PM_{2.5}$ levels and reached the
almost same value (about 4.8 ± 0.5) as those at WD and DBT when the concentrations of $PM_{2.5}$
were above 150 μg m$^{-3}$ (the serious pollution events). Because there were relatively sparse
emissions from vehicles and industries at WD and DBT, the almost constant of OC/EC ratios
under the different levels of $PM_{2.5}$ at WD and DBT further revealed that atmospheric OC and EC
were dominated by the local residential coal combustion. The almost same OC/EC ratios at the
four sampling sites with the concentrations of $PM_{2.5}$ greater than 150 μg m$^{-3}$ indicated that the
residential coal combustion also made dominant contribution to atmospheric OC and EC in the
two cities during the severe pollution period. Our previous study (C. Liu et al., 2016) also found
that the contribution from residential coal combustion to atmospheric VOCs increased from 23 %
to 33 % with increasing pollution levels in Beijing.
It should be mentioned that the OC/EC ratios observed at DBT and WD were about a factor of 2.7
less than that (13.1) of the emission from the residential coal combustion, whereas at BJ and BD
were too high to be explained by direct emissions from diesel (0.4-0.8) and gasoline (3.1) vehicles
(Shah et al., 2004; Geller et al., 2006). The OC emitted from the residential coal combustion might
be easily degraded or volatilized in the atmosphere, resulting in the relatively low OC/EC ratios



observed at DBT and WD. In China, aromatic compounds as typical pollutants from vehicle
emissions are very reactive to make contribution to secondary organic aerosols (SOA) (Zhang et
al., 2017), which was suspected to make evident contribution to the OC/EC ratios at BJ and BD
when the atmospheric EC concentrations were relatively low. For example, the extremely high
OC/EC ratios (> 6.0) at BJ and BD only occurred when the atmospheric EC concentrations were
less than 3.2 $\mu g\ m^{-3}$ at BJ and 5.4 $\mu g\ m^{-3}$ at BD. Because the atmospheric EC concentrations at BJ
and BD were about a factor of 4-6 greater during the serious pollution events than during the
slight pollution events, the effect of SOA formation on the OC/EC ratios would become less
during the serious pollution events if the SOA formation rate kept constant.
**3.4 Correlations among the species in PM$_{2.5}$**
The correlations among the WSIs, OC and EC in PM$_{2.5}$ at the four sampling sites are listed in
Table 4. The number of the species involved in significant correlations evidently increased from
the countryside to the cities and was 18, 28, 30 and 36 at DBT, WD, BD and BJ, respectively. The
significant correlations among the species could be classified as three types: 1) associated with OC
and EC; 2) associated with $Ca^{2+}$ and $Mg^{2+}$; and 3) associated with $K^+$. Three types of significant
correlations at DBT were independent of each other, whereas they were involved in interrelation
more and more from WD to BJ. The independence for the three types of significant correlations at
DBT further confirmed that residential coal combustion was preferentially dominant source for
atmospheric OC and EC. The significant correlations among OC, EC, $NO_3^-$, $NH_4^+$ and $Cl^-$ at DBT
indicated that the OC and EC emitted from the residential coal combustion could quickly
accelerate secondary formation of $NO_3^-$, $NH_4^+$ and $Cl^-$ via heterogeneous or multiphase reactions
of $NO_x$, $NH_3$ and HCl which have been verified to be emitted from the residential coal combustion





(Wang et al., 2005; Shapiro et al., 2007; Bläsing and Müller, 2010; Meng et al., 2011; Zhang et al.,
2013; Gao et al., 2015; Li et al., 2016a; Huang et al., 2016). The interrelation for the three types of
significant correlations at WD, BD and BJ implied that complex sources including local emissions
and regional transportation were dominant for atmospheric species in the cities. The species
associated with $Ca^{2+}$ and $Mg^{2+}$ from construction and road dust (Liang et al., 2016) as well as the
species associated with $K^+$ from biomass (municipal solid waste) burning (Gao et al., 2011; J. Li et
al., 2014; Yao et al., 2016) in the cities would accumulate under stagnant air conditions at the earth
surface, meanwhile the OC and EC concentrations could also increase due to the air parcel
transportation with abundant OC and EC in the upper layer from the south-southwest directions
(Fig. 8). It is interesting to note that the significant correlations among OC, EC, $NO_3^-$, $NH_4^+$ and
$Cl^-$ were found at the four sampling sites, whereas the significant correlation between OC (or EC)
and $SO_4^{2-}$ was only found at BJ. Because the sampling sites of DBT, WD and BD are close to the
source of OC and EC from the residential coal combustion, the significant correlations among OC,
EC, $NO_3^-$, $NH_4^+$ and $Cl^-$ but the insignificant correlation between OC (or EC) and $SO_4^{2-}$ implied
that the formation rate of $SO_4^{2-}$ via heterogeneous or multiphase reactions might be relatively
slower than those of $NO_3^-$, $NH_4^+$ and $Cl^-$. The OC, EC and $SO_2$ emitted from the residential coal
combustion experienced the relatively long period of excursion to be transported to Beijing,
resulting in the significant correlation between OC (or EC) and $SO_4^{2-}$ at BJ.
As listed in Table 5, the pronounced correlations for [As] vs. [Se] and [Cu] vs. [Zn] at the four
sampling sites indicated that the two pairs of elements were from the common sources. Based on
the remarkable elevations of As and Se near a coal-fired power plant with respect to the
background site, Jayasekher (2009) pointed out that their significant correlation can be used as the



tracer for coal combustion. Because Cu and Zn have been found to be mainly released from the
additives of vehicle lubricating oils, brake and tire wear during transportation activities (Yu et al.,
2013; Zhang et al., 2013; Tan et al., 2016), their significant correlation has been used as the tracer
for vehicle emissions. Both coal combustion and vehicle emissions could make contribution to
atmospheric Pb (Zhang et al., 2013; Gao et al., 2016), and thus the correlations for [Pb] vs.
[Cu+Zn] and [Pb] vs. [As+Se] could reflect their local dominant sources. As shown in Fig. 10, the
significant correlation between [Pb] and [Cu+Zn] but no correlation between [Pb] and [As+Se]
were found at BJ, whereas the correlations at the rural site of DBT were on the contrary, indicating
that atmospheric Pb, Cu and Zn at BJ were mainly related to the vehicle emissions and
atmospheric Pb, As and Se at DBT were dominated by residential coal combustion. Because the
sampling sites of BD and WD were affected by both vehicle emissions and residential coal
combustion, the significant correlations between [Pb] and [Cu+Zn] as well as [Pb] and [As+Se]
were found at the two sampling sites. Although there was no correlation between [Pb] and [As+Se]
at BJ, the contribution of residential coal combustion to atmospheric $PM_{2.5}$ in the city of BJ could
not be excluded because the trace elements from coal combustion are mainly present in relatively
large particles (0.8-2.5 μm) which might quickly deposit near their sources (Wang et al., 2008).
**3.5 Source apportionment of $PM_{2.5}$ at the four sampling sites**
The source characteristics of $PM_{2.5}$ at the four sampling sites were analyzed by the CMC method
which has been described in detail in section 2.3. The average proportions of the species from
different sources in $PM_{2.5}$ during the sampling period at the four sampling sites are comparatively
shown in Fig. 11. It is evident that secondary aerosols (SIA + SOC) accounted for the largest
proportion (about 32-41 %) in $PM_{2.5}$, followed by POC (about 24-28 %), EC (about 6-8 %),



mineral dust (about 2-8 %) and $Cl^-_{cc}$ (about 2-5 %) at the four sampling sites. The proportion of
mineral dust was the highest at BJ and the lowest at DBT among the four sampling sites, whereas
the proportion of $Cl^-_{cc}$ was on the contrary. Because the concentrations of the mineral dust
compounds were much higher under stagnant weather condition than under clean days at BJ, the
remarkably high proportion of mineral dust at BJ was mainly ascribed to the emissions from road
dust and construction (Liang et al., 2016) during the sampling period. The obviously high
proportion of $Cl^-_{cc}$ at DBT was ascribed to the emission from residential coal combustion (Shen et
al., 2016). In addition, the proportions of TEO, $K^+_{bb}$ and $Cl^-_{ss}$ were less than about 2 %, which
were insignificant to the sources of $PM_{2.5}$ at the four sampling sites during the sampling period.
Atmospheric Primary Organic Matters (POM) and $Cl^-_{cc}$ at the four sampling sites could be
estimated based on POM≈POC × 1.6 (Cheung et al., 2005; Hsu et al., 2010b; Han et al., 2015)
and the formulas (1)-(4), respectively. The sum of POM, EC and $Cl^-_{cc}$ at DBT was assumed to be
solely from residential coal combustion, accounting for about 58% in $PM_{2.5}$ (Fig. 12). Assuming
that the ratio of $Cl^-_{cc}$ to the sum of POM, EC and $Cl^-_{cc}$ was constant for coal combustion at the four
sampling sites, the primary contribution of coal combustion to atmospheric $PM_{2.5}$ at BJ, BD and
WD could be estimated to be 32 %, 49 % and 43 % (Fig. 12), respectively. The annual residential
coal consumption mainly focused on the four months in winter, accounting for about 11 % of the
total coal consumption in the region of BTH. Because the emission factor of $PM_{2.5}$ from
residential coal combustion (about 1054-12910 mg $kg^{-1}$) was about 1-3 orders of magnitude
greater than those from industry boilers or coal power plants (about 16-100 mg $kg^{-1}$) (Chen et al.,
2005; Zhang et al., 2008), the estimated proportions of the contribution of coal combustion to
atmospheric $PM_{2.5}$ at the four sampling sites during the winter were mainly ascribed to residential




coal combustion. If only the primary $PM_{2.5}$ was considered, the contribution of residential coal
combustion to the primary $PM_{2.5}$ at BJ would achieve to be about 59 % which was in line with the
value of 57 % estimated by J. Liu et al. (2016) for the winter of 2010 in Beijing.
**4 Conclusions**
Based on the comprehensive analysis of the levels, composition characteristics, the correlations of
the key species in $PM_{2.5}$ and the back trajectories, residential coal combustion in the North China
during winter was found not only to be the dominant source for atmospheric OC, EC, $Cl^-$, $NO_3^-$,
$SO_4^{2-}$ and $NH_4^+$ in rural areas but also to make evident contribution to the species in cities.
According to the CMC method, the contributions of the primary particle emission from residential
coal combustion to atmospheric $PM_{2.5}$ at BJ, BD, WD and DBT during winter were estimated to
be 32 %, 49 %, 43 % and 58 %, respectively. Therefore, strict control measures should be
implemented for the emissions from residential coal combustion to mitigate the currently serious
$PM_{2.5}$ pollution during the winter in the North China.
**Author contribution**
**Y. J. Mu** designed the experiments and prepared the manuscript. **P. F. Liu** carried out the
experiments and prepared the manuscript. **C. Y. Xue** and **C. L. Zhang** carried out the experiments.
**J. F. Liu**, **Y. Y. Zhang**, **D. Tian** and **C. Ye** were involved in part of the work. **H. X. Zhang**
provided the meteorological data and trace gases in Beijing. **J. Guan** provided the meteorological
data and trace gases in Baoding and Wangdu.
**Acknowledgements**
This work was supported by the National Natural Science Foundation of China (No. 21477142,
91544211 and 41575121), the Special Fund for Environmental Research in the Public Interest (No.



201509002) and the projects of the Strategic Priority Research Program (B) of the Chinese
Academy of Sciences (No. XDB05010100).

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

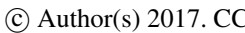


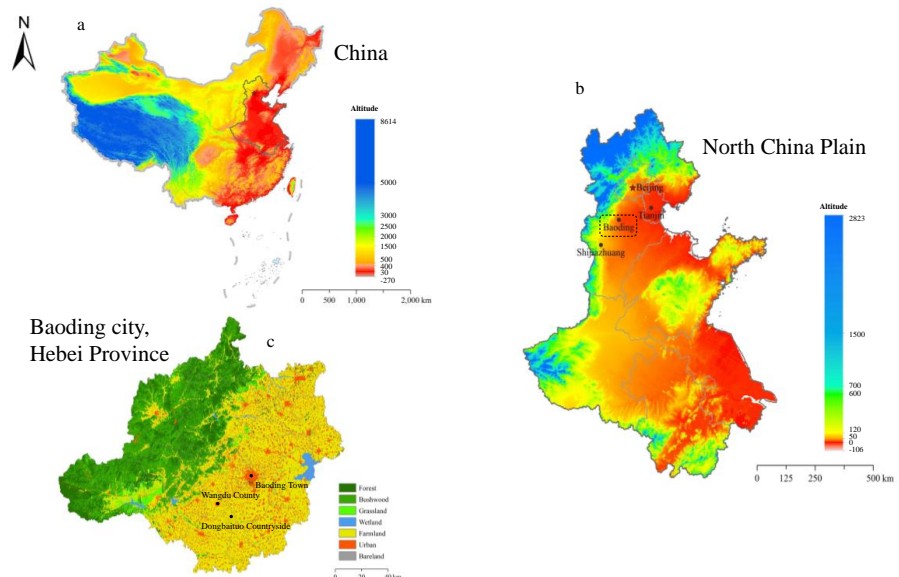


**Figure 1.** China (a), the North China Plain (b) and Baoding city in Hebei Province (c). The
locations of sampling sites (BJ, BD, WD and DBT) as well as Tianjin municipality and
Shijiazhuang as provincial capital of Hebei are marked.

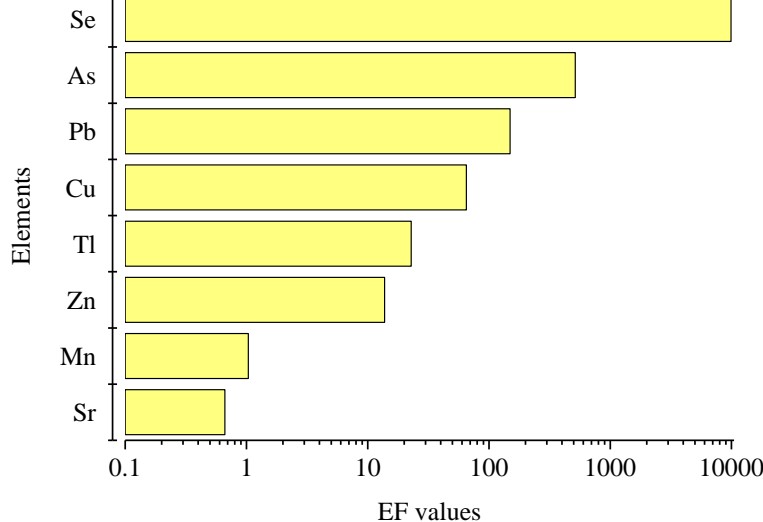


**Figure 2.** Enrichment factor values for trace elements in $PM_{2.5}$.





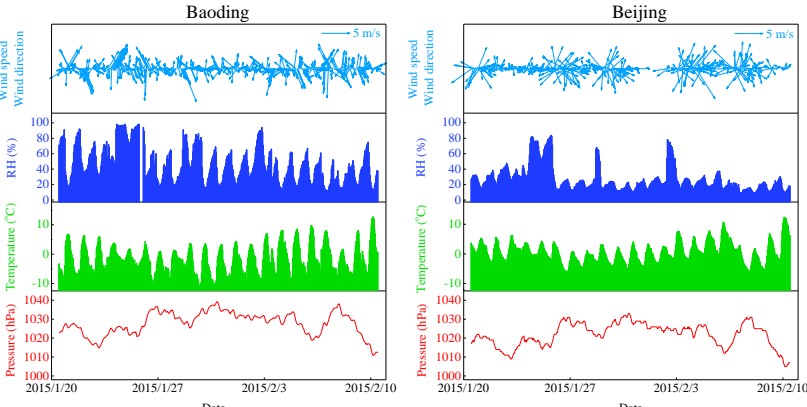


**Figure 3.** The wind speed, wind direction, RH, temperature and barometric pressure at BD and BJ
during the sampling period in the winter of 2015.

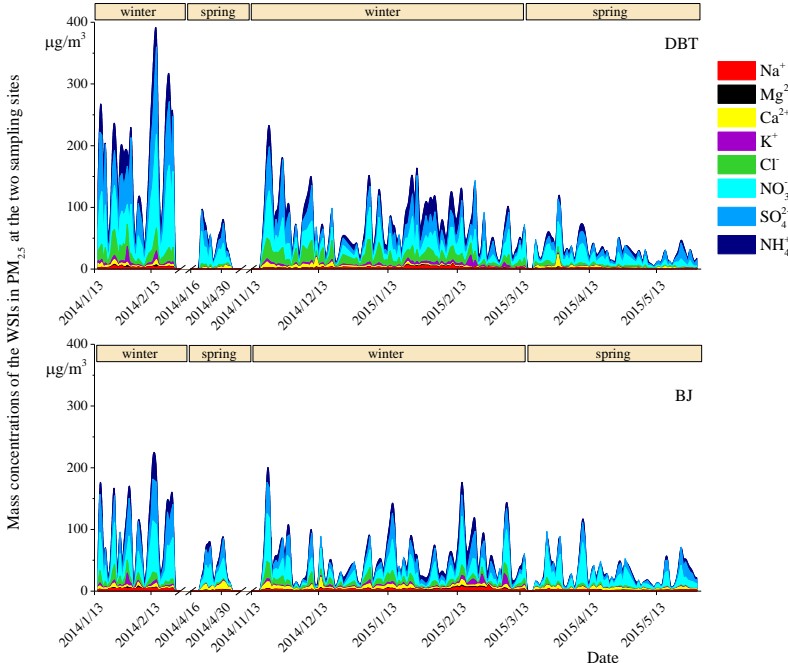


**Figure 4.** The mass concentrations of the WSIs in PM$_{2.5}$ at DBT and BJ during the sampling
period in the winters and springs of 2014-2015.



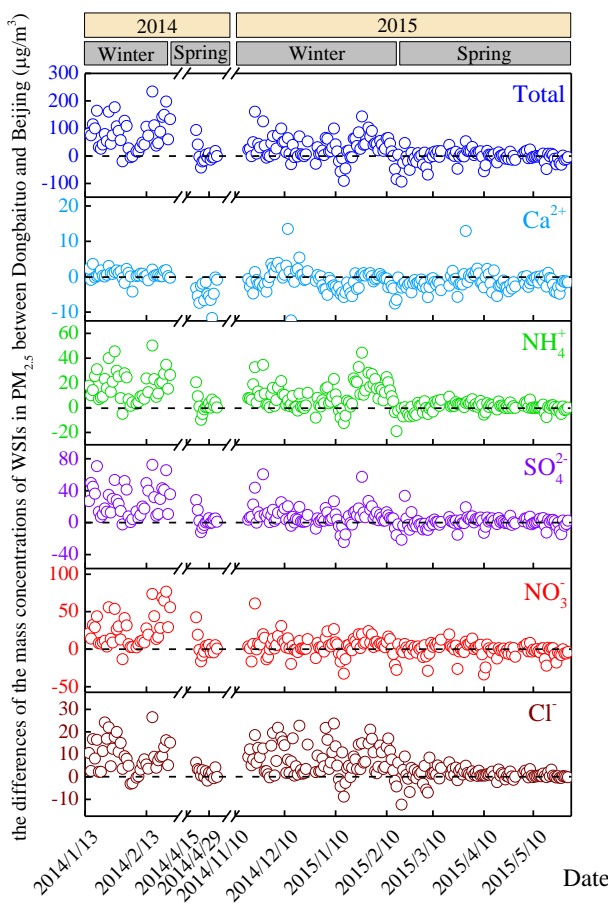


**Figure 5.** The D-values of the mass concentrations of WSIs in PM$_{2.5}$ between DBT and BJ during

the sampling period in the winters and springs of 2014-2015.





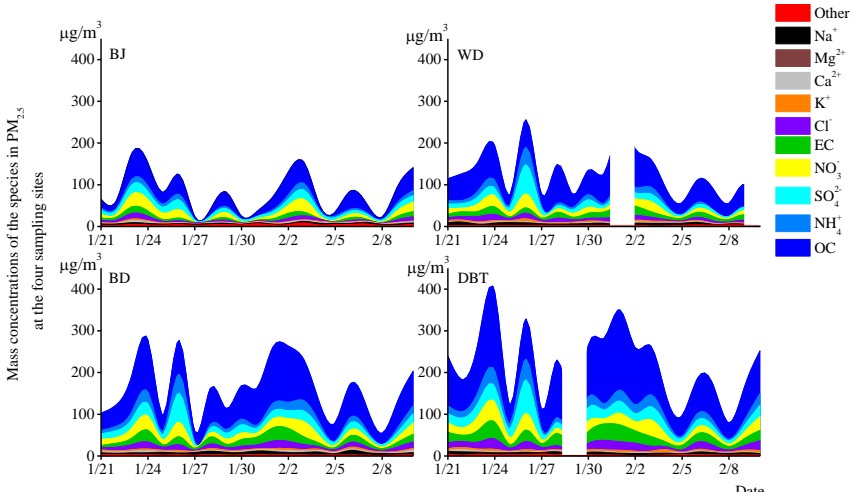


**Figure 6.** Daily variation of the species in PM$_{2.5}$ at the four sampling sites during the sampling
period in the winter of 2015.

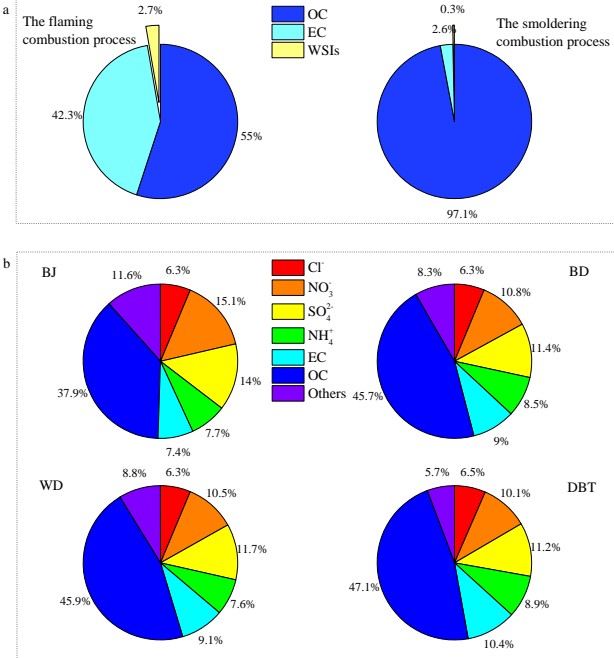


**Figure 7.** The mass proportions of OC, EC and WSIs from residential coal combustion under the
flaming and smoldering combustion processes (a), and the average mass proportions of the typical





species in PM$_{2.5}$ at the four sampling sites during the sampling period in the winter of 2015 (b).

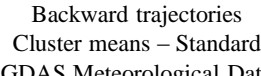

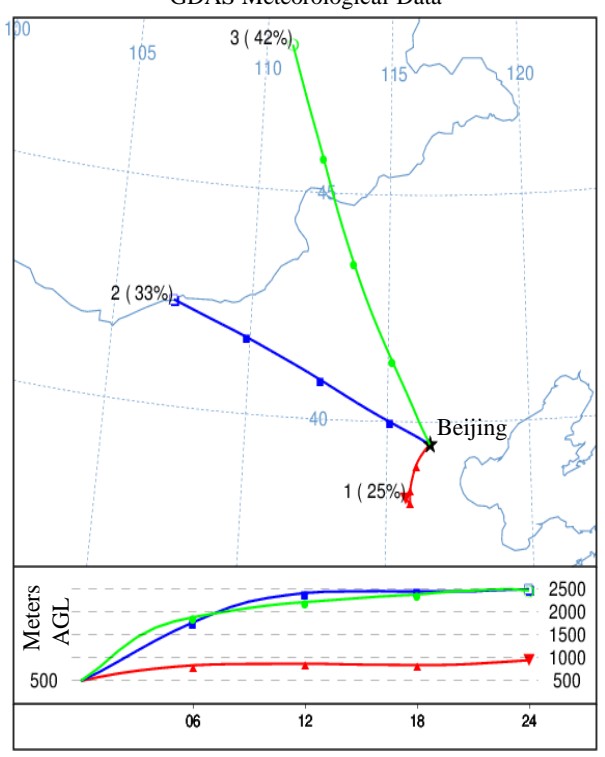

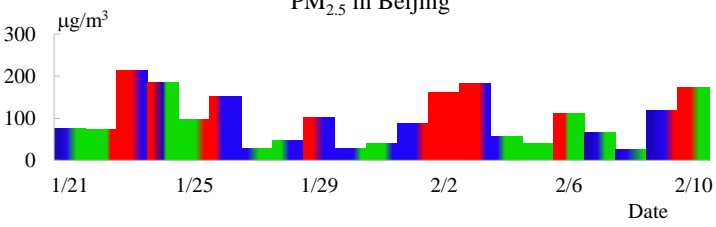


**Figure 8.** The back trajectory cluster analysis and the corresponding PM$_{2.5}$ concentrations in

793          Beijing during the sampling period in the winter of 2015.






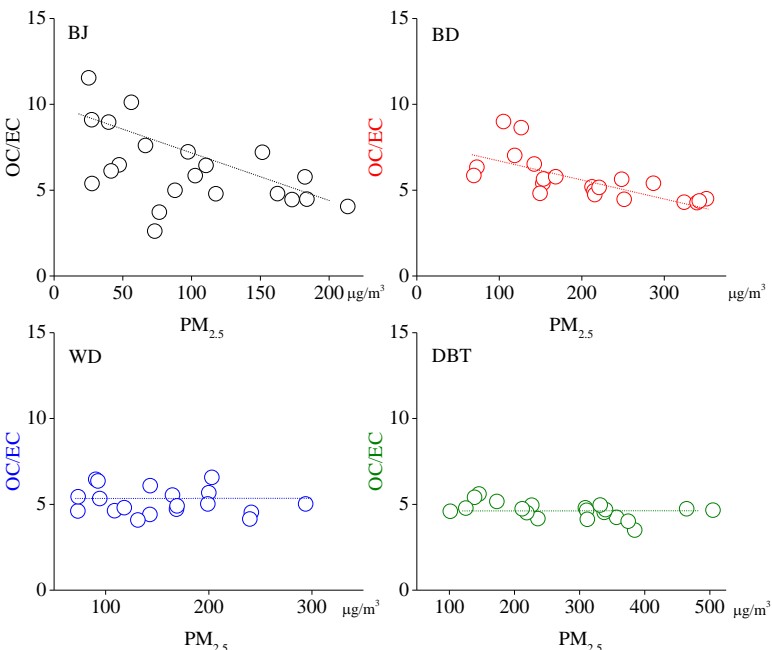


**Figure 9.** The correlations between the OC/EC ratios and the PM$_{2.5}$ concentrations at the four

797                     sampling sites during the sampling period in the winter of 2015.

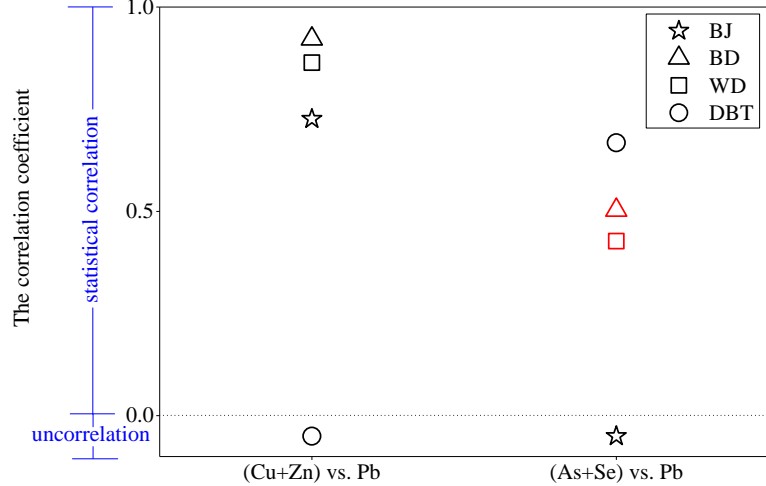


**Figure 10.** The statistical correlations for [Cu+Zn] vs. [Pb] and [As+Se] vs. [Pb] in PM$_{2.5}$ at the
four sampling sites during the sampling period in the winter of 2015. The uncorrelated results are



also marked below zero of Y axis. The red and black symbols represent for $p < 0.05$ and $p < 0.01$,

802                      respectively.

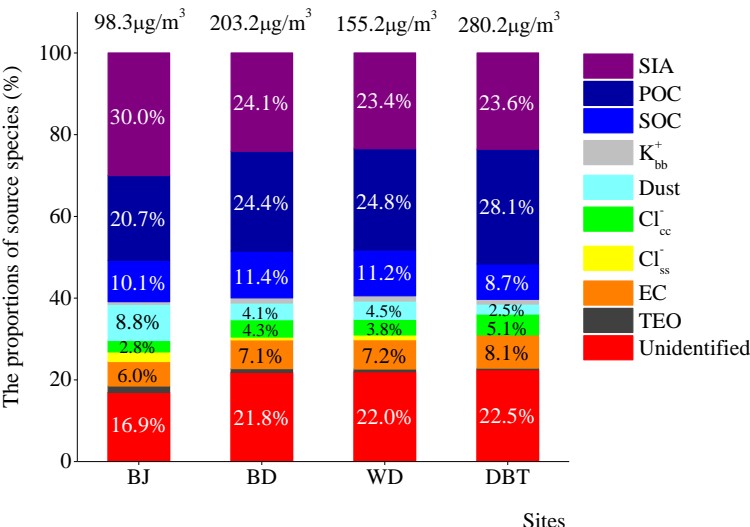


**Figure 11.** The proportions of source species under the constructed chemical mass closures for PM$_{2.5}$ at the four sampling sites during the sampling period in the winter of 2015. Average mass concentrations of PM$_{2.5}$ at each sampling site, including all of source species and unidentified fractions, are also marked at the top of bar charts.

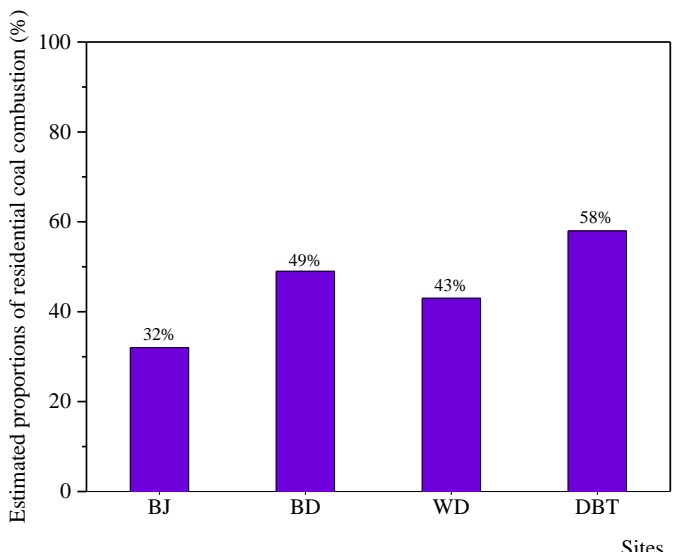






**Figure 12.** The estimated contributions of coal combustion to the PM$_{2.5}$ at the four sampling sites
during the sampling period in the winter of 2015.

**Table 1.** The average mass concentrations of WSIs in PM$_{2.5}$ at DBT and BJ during the sampling
period in the winters and springs of 2014-2015 ($\mu g\ m^{-3}$).

| WSIs | spring | | winter | |
|---|---|---|---|---|
| | DBT | BJ | DBT | BJ |
| Na$^+$ | 1.0 ±0.5 | 1.4 ±0.5 | 2.4 ±1.3 | 3.1 ±1.4 |
| Mg$^{2+}$ | 0.2 ±0.2 | 0.3 ±0.2 | 0.7 ±0.5 | 0.8 ±0.7 |
| Ca$^{2+}$ | 1.7 ±2.4 | 3.4 ±2.5 | 2.6 ±2.1 | 3.4 ±2.3 |
| K$^+$ | 0.5 ±0.5 | 0.7 ±0.4 | 3.2 ±3.0 | 3.0 ±6.0 |
| NH$_4^+$ | 6.1 ±5.1 | 4.8 ±4.7 | 23.1 ±17.9 | 13.2 ±11.6 |
| NO$_3^-$ | 12.5 ±11.2 | 13.6 ±13.2 | 28.4 ±28.0 | 19.0 ±20.0 |
| SO$_4^{2-}$ | 10.5 ±8.2 | 9.2 ±8.6 | 29.0 ±28.1 | 17.4 ±16.5 |
| Cl$^-$ | 2.9 ±2.2 | 1.8 ±1.6 | 14.1 ±9.4 | 7.2 ±6.0 |
| Total | 35.3 ±26.7 | 35.1 ±28.7 | 103.3 ±81.3 | 67.0 ±55.2 |


**Table 2.** The average mass concentrations (Mean ±SD) of PM$_{2.5}$ species, NO$_2$ and SO$_2$ at the four
sampling sites during the sampling period in the winter of 2015 ($\mu g\ m^{-3}$).

| Species | BJ | BD | WD | DBT |
|---|---|---|---|---|
| Na$^+$ | 2.5 ±0.7 | 4.8 ±2.0 | 4.5 ±1.7 | 4.3 ±1.2 |
| Mg$^{2+}$ | 0.3 ±0.1 | 0.4 ±0.1 | 0.3 ±0.1 | 0.4 ±0.2 |
| Ca$^{2+}$ | 1.8 ±0.9 | 2.6 ±0.8 | 1.7 ±0.6 | 2.0 ±0.8 |
| K$^+$ | 0.7 ±0.8 | 2.5 ±1.0 | 2.0 ±1.4 | 3.1 ±1.3 |
| NH$_4^+$ | 6.0 ±5.0 | 13.3 ±11.0 | 9.3 ±9.5 | 18.7 ±11.7 |
| NO$_3^-$ | 11.7 ±10.1 | 16.6 ±10.3 | 13.0 ±8.2 | 21.0 ±12.2 |
| SO$_4^{2-}$ | 11.2 ±6.5 | 18.1 ±14.1 | 14.5 ±14.5 | 24.1 ±16.1 |
| Cl$^-$ | 5.0 ±3.6 | 9.5 ±4.2 | 7.8 ±3.5 | 13.4 ±6.0 |
| OC | 28.6 ±19.6 | 70.2 ±31.2 | 57.2 ±21.3 | 100.0 ±42.9 |
| EC | 5.5 ±4.5 | 13.5 ±7.8 | 11.4 ±4.7 | 21.6 ±10.2 |
| Al | 0.6 ±0.8 | 0.6 ±0.1 | 0.5 ±0.2 | 0.5 ±0.1 |
| Mn | 0.1 ±0.1 | 0.1 ±0.1 | 0.1 ±0.1 | 0.2 ±0.3 |
| Fe | 2.1 ±0.8 | 0.6 ±0.2 | 0.8 ±0.6 | 1.3 ±0.6 |
| Cu | 0.6 ±0.3 | 0.3 ±0.1 | 0.2 ±0.1 | 0.1 ±0.1 |
| Zn | 0.1 ±0.1 | 0.2 ±0.1 | 0.1 ±0.1 | 0.1 ±0.1 |
| As | 0.1 ±0.1 | 0.3 ±0.1 | 0.2 ±0.1 | 0.1 ±0.1 |
| Se | 0.1 ±0.0 | 0.1 ±0.1 | 0.1 ±0.0 | 0.1 ±0.0 |
| Sr | 0.0 ±0.0 | 0.1 ±0.0 | 0.0 ±0.0 | 0.0 ±0.0 |
| Tl | 0.0 ±0.0 | 0.0 ±0.0 | 0.0 ±0.0 | 0.0 ±0.0 |
| Pb | 0.2 ±0.2 | 0.4 ±0.3 | 0.2 ±0.1 | 0.3 ±0.1 |
| The total | 80.1 ±47.7 | 159.5 ±70.3 | 121.7 ±51.8 | 218.4 ±87.1 |
| NO$_2$ | 36.5 ±17.4 | 60.4 ±23.4 | 76.1 ±19.2 | - |
| SO$_2$ | 63.9 ±31.7 | 181.7 ±62.4 | 101.3 ±39.4 | - |






**Table 3.** The emission factors (Mean ± SD) (g kg$^{-1}$ coal) of OC and EC from residential coal combustion during the flaming combustion process, the smoldering combustion process and the entire combustion process.

| Emission factors | the flaming combustion process | the smoldering combustion process | the entire combustion process |
|---|---|---|---|
| OC | 1.83 ± 1.19 | 17.11 ± 0.79 | 10.99 ± 0.95 |
| EC | 1.40 ± 0.11 | 0.46 ± 0.03 | 0.84 ± 0.06 |

**Table 4.** The correlations of several typical species in PM$_{2.5}$ at the four sampling sites during the sampling period in the winter of 2015.

| n=21 | BJ | | | | | | | | |
|---|---|---|---|---|---|---|---|---|---|
| | Mg$^{2+}$ | Ca$^{2+}$ | K$^+$ | Cl$^-$ | NO$_3^-$ | SO$_4^{2-}$ | NH$_4^+$ | OC | EC |
| Mg$^{2+}$ | 1 | | | | | | | | |
| Ca$^{2+}$ | 0.895** | 1 | | | | | | | |
| K$^+$ | 0.634** | 0.862** | 1 | | | | | | |
| Cl$^-$ | 0.856** | 0.899** | 0.791** | 1 | | | | | |
| NO$_3^-$ | 0.803** | 0.768** | 0.637** | 0.905** | 1 | | | | |
| SO$_4^{2-}$ | 0.679** | 0.660** | 0.590** | 0.804** | 0.950** | 1 | | | |
| NH$_4^+$ | 0.718** | 0.667** | 0.543* | 0.834** | 0.971** | 0.959** | 1 | | |
| OC | 0.845** | 0.751** | 0.560** | 0.848** | 0.919** | 0.838** | 0.895** | 1 | |
| EC | 0.849** | 0.851** | 0.679** | 0.932** | 0.877** | 0.769** | 0.823** | 0.936** | 1 |

| n=21 | BD | | | | | | | | |
|---|---|---|---|---|---|---|---|---|---|
| | Mg$^{2+}$ | Ca$^{2+}$ | K$^+$ | Cl$^-$ | NO$_3^-$ | SO$_4^{2-}$ | NH$_4^+$ | OC | EC |
| Mg$^{2+}$ | 1 | | | | | | | | |
| Ca$^{2+}$ | 0.805** | 1 | | | | | | | |
| K$^+$ | 0.697** | 0.556** | 1 | | | | | | |
| Cl$^-$ | 0.714** | 0.659** | 0.789** | 1 | | | | | |
| NO$_3^-$ | 0.554** | 0.560** | 0.675** | 0.757** | 1 | | | | |
| SO$_4^{2-}$ | 0.022 | 0.107 | 0.491* | 0.499* | 0.764** | 1 | | | |
| NH$_4^+$ | 0.315 | 0.331 | 0.659** | 0.721** | 0.920** | 0.941** | 1 | | |
| OC | 0.743** | 0.576** | 0.705** | 0.936** | 0.674** | 0.369 | 0.614** | 1 | |
| EC | 0.698** | 0.560** | 0.702** | 0.939** | 0.660** | 0.410 | 0.633** | 0.984** | 1 |

| n=19 | WD | | | | | | | | |
|---|---|---|---|---|---|---|---|---|---|
| | Mg$^{2+}$ | Ca$^{2+}$ | K$^+$ | Cl$^-$ | NO$_3^-$ | SO$_4^{2-}$ | NH$_4^+$ | OC | EC |
| Mg$^{2+}$ | 1 | | | | | | | | |
| Ca$^{2+}$ | 0.897** | 1 | | | | | | | |
| K$^+$ | 0.226 | 0.457* | 1 | | | | | | |
| Cl$^-$ | 0.532* | 0.663** | 0.598** | 1 | | | | | |
| NO$_3^-$ | 0.468* | 0.677** | 0.712** | 0.796** | 1 | | | | |
| SO$_4^{2-}$ | 0.097 | 0.358 | 0.874** | 0.552* | 0.770** | 1 | | | |
| NH$_4^+$ | 0.306 | 0.563** | 0.906** | 0.735** | 0.901** | 0.945** | 1 | | |
| OC | 0.463* | 0.543* | 0.372 | 0.816** | 0.471* | 0.222 | 0.581* | 1 | |
| EC | 0.553* | 0.638** | 0.339 | 0.763** | 0.510* | 0.214 | 0.565* | 0.925** | 1 |





| n=20 | DBT | | | | | | | | |
|---|---|---|---|---|---|---|---|---|---|
| | $Mg^{2+}$ | $Ca^{2+}$ | $K^+$ | $Cl^-$ | $NO_3^-$ | $SO_4^{2-}$ | $NH_4^+$ | OC | EC |
| $Mg^{2+}$ | 1 | | | | | | | | |
| $Ca^{2+}$ | 0.721** | 1 | | | | | | | |
| $K^+$ | 0.191 | 0.407 | 1 | | | | | | |
| $Cl^-$ | -0.061 | 0.316 | 0.519* | 1 | | | | | |
| $NO_3^-$ | -0.241 | 0.161 | 0.579** | 0.642** | 1 | | | | |
| $SO_4^{2-}$ | -0.133 | 0.109 | 0.458* | 0.482* | 0.744** | 1 | | | |
| $NH_4^+$ | -0.223 | 0.125 | 0.558* | 0.697** | 0.928** | 0.914** | 1 | | |
| OC | 0.067 | 0.159 | 0.419 | 0.772** | 0.570** | 0.293 | 0.557* | 1 | |
| EC | 0.051 | 0.169 | 0.419 | 0.838** | 0.585** | 0.400 | 0.624** | 0.977** | 1 |

*, ** represent for $p < 0.05$ and $p < 0.01$, respectively.

**Table 5.** The correlations between [Zn] vs. [Cu] and [As] vs. [Se] in $PM_{2.5}$ at the four sampling
sites during the sampling period in the winter of 2015.

| Elements | BJ (n=21) | BD (n=21) | WD (n=19) | DBT (n=20) |
|---|---|---|---|---|
| [Zn] vs. [Cu] | 0.607** | 0.479* | 0.620* | 0.659** |
| [As] vs. [Se] | 0.662** | 0.664** | 0.959** | 0.871** |

*, ** represent for $p < 0.05$ and $p < 0.01$, respectively.


















