# Peer review of "The contribution of residential coal combustion to atmospheric PM2.5"

_Atmospheric Chemistry and Physics, 2017_

## Referee Comment (RC1) · Anonymous Referee #2 · 23 May 2017

**Comments on "The contribution of residential coal combustion to atmospheric PM$_{2.5}$ in the North China during winter"**

The air pollution is very serious in China especially the North China during winter. Moreover, the rapid formation of PM$_{2.5}$ is more frequency, and the reason of this phenomenon is not very clearly. This article studied the concentration, composition, and the correlations of the key species of PM$_{2.5}$ in four sampling sites of North China. The study demonstrated that the residential coal combustion was dominant source of atmospheric OC, EC, Cl$^-$, NO$_3^-$, SO$_4^{2-}$ and NH$_4^+$ in both rural areas and cities in the four sites of North China. The author also used the CMC method to calculate the contributions of the primary particle emission form residential coal combustion to PM$_{2.5}$ at the four sites during winter. The article is suitable to be published in this Journal. I recommended it to be accepted after minor revision.

Comments:

1. 361 line: The author demonstrated that the formation rate of SO$_4^{2-}$ via heterogeneous or multiphase reactions might be slower, so the correlation between OC (or EC) and SO$_4^{2-}$ was insignificant. This conclusion is not convinced. The author should give more evidence to support this point, such as some analyzing of trace gases (SO$_2$, CO, NOx and so on) during the period in four sites or reference some laboratory studies on the formation rate of SO$_4^{2-}$ via heterogeneous or multiphase reactions.

2. There are some grammar error, the author should improve the English of the article.

---

## Referee Comment (RC2) · Anonymous Referee #1 · 26 Jul 2017

This manuscript by Liu et al. reports PM2.5 and its major components in winter and spring seasons at four sites in North China. Chemical compositions and spatial difference are discussed. The major sources are also attributed. Generally, the study is well-designed and the manuscript should be published after my concerns are addressed.

Major concerns 1. Line 144-159. The authors claim that Cl/Na ratio is 1.4 in coal combustion. And if the ratio high than 1.4, atmospheric Cl and Na would be considered to be totally from coal combustion. In fact, biomass burning, including wild fires, open straw burning and biofuel combustion also emits Na and Cl with the Cl/Na ratios of 1-6

(e.g. Schauer ES&T 2001). Moreover, the OC/EC ratios in biomass burning samples are as high as those in coal combustion (Table 3). Since biofuel combustion for heating is also enhanced during winter in the northern China, why and how did the authors rule out the influence of biomass burning on PM2.5 at the four sites? 2. The authors discuss the spatial difference of PM2.5 and the major components at the four sites. Did the meteorological conditions, such as planetary boundary layer (PBL), cause such a spatial difference? 3. SOC is estimated by the EC-tracer OC/EC method. However, previous studies have demonstrated that this method could overestimate SOC under the influence of coal combustion and biomass burning, especially in wintertime. As discussed in the manuscript, coal combustion (maybe also biofuel combustion) had significant impact on PM at the sampling sites in wintertime. Thus, the concentrations and mass fractions of SOC in the winter (Figure 11) should be overestimate.

Specific comments 1. The citation need to be re-formatted throughout the main text. For example, in line 48, the citation formats for the two references are different (Huang et al., 2014; P. Liu et al., 2016). 2. Line 230-232. Is the difference in concentrations statistically significant? Please add p-value to show the significance of the observed difference.

---

## Author Comment (AC1) · 3 Aug 2017

**A point-by-point response to the reviews**

Thank you for your valuable comments. The followings are our responses to your comments.

**Response to Reviewer #2**

**Comment 1:** The air pollution is very serious in China especially the North China during winter. Moreover, the rapid formation of $PM_{2.5}$ is more frequency, and the reason of this phenomenon is not very clearly. This article studied the concentration, composition, and the correlations of the key species of $PM_{2.5}$ in four sampling sites of North China. The study demonstrated that the residential coal combustion was dominant source of atmospheric OC, EC, $Cl^-$, $NO_3^-$, $SO_4^{2-}$ and $NH_4^+$ in both rural areas and cities in the four sites of North China. The author also used the CMC method to calculate the contributions of the primary particle emission from residential coal combustion to $PM_{2.5}$ at the four sites during winter. The article is suitable to be published in this Journal. I recommended it to be accepted after minor revision.

**Answer:** Thank you for your positive evaluation of our work. The followings are our responses to your comments.

**Comment 2:** line 361: The author demonstrated that the formation rate of $SO_4^{2-}$ via heterogeneous or multiphase reactions might be slower, so the correlation between OC (or EC) and $SO_4^{2-}$ was insignificant. This conclusion is not convinced. The author should give more evidence to support this point, such as some analyzing of trace gases ($SO_2$, CO, $NO_x$ and so on) during the period in four sites or reference some laboratory studies on the formation rate of $SO_4^{2-}$ via heterogeneous or multiphase reactions.

**Answer:** Thank you for your valuable suggestion. The reactive uptake coefficients of $SO_2$ oxidation by $O_3$ were reported to be from $4.3 \times 10^{-8}$ to $7 \times 10^{-7}$ on different mineral aerosols and from $1 \times 10^{-6}$ to $6 \times 10^{-6}$ on soot particles (Wu et al., 2011; Song et al., 2012), which were at least one order of magnitude less than those of $NO_2$ ($1.03 \times 10^{-2}$-$3.43 \times 10^{-3}$ on soot particles and $1.03 \times 10^{-6}$-$1.2 \times 10^{-5}$ on mineral aerosols) (Underwood et al., 2001; Esteve et al., 2004; Ma et al., 2011; Ma et al., 2017). The information has been added in the revised manuscript.

**Comment 3:** There are some grammar error, the author should improve the English of the article.

**Answer:** According to your suggestion, the English of the manuscript has been improved through carefully correcting grammar errors, which has been marked in the revised manuscript.

**References**

Esteve, W., Budzinski, H., and Villenave, E.: Relative rate constants for the heterogeneous reactions of OH, $NO_2$ and NO radicals with polycyclic aromatic hydrocarbons adsorbed on carbonaceous particles. Part 1: PAHs adsorbed on 1–2μm calibrated graphite particles, Atmospheric Environment, 38, 6063-6072, 10.1016/j.atmosenv.2004.05.059, 2004.

Ma, J., Liu, Y., and He, H.: Heterogeneous reactions between $NO_2$ and anthracene adsorbed on $SiO_2$ and MgO, Atmospheric Environment, 45, 917-924, 10.1016/j.atmosenv.2010.11.012, 2011.

Ma, Q., Wang, T., Liu, C., He, H., Wang, Z., Wang, W., and Liang, Y.: $SO_2$ Initiates the Efficient Conversion of $NO_2$ to HONO on MgO Surface, Environ. Sci. Technol., 51, 3767-3775, 10.1021/acs.est.6b05724, 2017.

Song, H., Shang, J., Zhu, T., Zhao, L., and Ye, J. H.: Heterogeneous Oxidation of $SO_2$ by Ozone on the Surface of Black Carbon Particles, Chem. J. Chin. Univ. Chin., 33, 2295-2302, 10.7503/cjcu20120024, 2012.

Underwood, G. M., Song, C. H., Phadnis, M., Carmichael, G. R., and Grassian, V. H.: Heterogeneous reactions of $NO_2$ and $HNO_3$ on oxides and mineral dust: A combined laboratory and modeling study, Journal of Geophysical Research: Atmospheres, 106, 18055-18066, 10.1029/2000jd900552, 2001.

Wu, L. Y., Tong, S. R., Wang, W. G., and Ge, M. F.: Effects of temperature on the heterogeneous oxidation of sulfur dioxide by ozone on calcium carbonate, Atmospheric Chemistry and Physics, 11, 6593-6605, 10.5194/acp-11-6593-2011, 2011.

---

## Author Comment (AC2) · 3 Aug 2017

A point-by-point response to the reviews

Thank you for your valuable comments. The followings are our responses to your comments.

**Response to Reviewer #1**
**Comment 1:** This manuscript by Liu et al. reports $PM_{2.5}$ and its major components in winter and spring seasons at four sites in North China. Chemical compositions and spatial difference are discussed. The major sources are also attributed. Generally, the study is well-designed and the manuscript should be published after my concerns are addressed.

**Answer:** Thank you for your positive evaluation of our work. The followings are our responses to your comments.

**Comment 2:** Line 144-159. The authors claim that Cl/Na ratio is 1.4 in coal combustion. And if the ratio high than 1.4, atmospheric Cl and Na would be considered to be totally from coal combustion. In fact, biomass burning, including wild fires, open straw burning and biofuel combustion also emits Na and Cl with the Cl/Na ratios of 1-6 (e.g. Schauer ES&T 2001). Moreover, the OC/EC ratios in biomass burning samples are as high as those in coal combustion (Table 3). Since biofuel combustion for heating is also enhanced during winter in the northern China, why and how did the authors rule out the influence of biomass burning on $PM_{2.5}$ at the four sites?

**Answer:** Yes, you are right. Biomass burning is indeed an important source for atmospheric $Na^+$ and $Cl^-$, with the $Cl^-/Na^+$ ratios of 1-6 (Schauer et al., 2001). However, biomass burning in the NCP region is mainly focusing on the harvest seasons in summer and autumn (Zong et al., 2016), and few farmers are currently combusting crop straws for household cooking and heating because of the inconvenience of biomass with respect to coal and liquid gas.
The ratios of OC and EC to $K^+$ from biomass burning had been measured to be about 3.9 and 0.8, respectively (Li et al., 2007; Yao et al., 2016), which was about one order of magnitude less than those (34.0 ± 9.3 for OC/ $K^+$ and 6.9 ± 1.7 for EC/ $K^+$) measured in this study. Assuming the atmospheric $K^+$ measured in this study was totally from biomass burning, the contribution of biomass burning to atmospheric carbonaceous aerosols could be roughly estimated to be only 2.8%-5.2% in $PM_{2.5}$ based on the typical ratios of OC and EC to $K^+$ from biomass burning and the mass proportions of $K^+$ (0.6%-1.1%, Fig. 11). Therefore, biomass burning during the sampling period in this study made minor contribution to atmospheric $PM_{2.5}$. According to your pertinent comment, the corresponding paragraph has been rephrased in the revised manuscript.

**Comment 3:** The authors discuss the spatial difference of $PM_{2.5}$ and the major components at the four sites. Did the meteorological conditions, such as planetary boundary layer (PBL), cause such a spatial difference?

**Answer:** The meteorological conditions, especially wind speed and planetary boundary layer (PBL), play pivotal roles in the dispersion and accumulation of atmospheric pollutants, which can cause spatial and temporal difference of pollutants. As for the sampling sites of BD, WD and DBT, the meteorological conditions could be considered as the same because of the short distances (< 36 km)

among them, and hence the spatial difference of $PM_{2.5}$ and the major components at the three sampling sites was rationally ascribed to the different source strengths. Although the distance between the sampling sites of BJ and BD is about 156 km, there was no significant difference of the wind speeds between the two sampling sites during the sampling period (1.4 ± 1.4 m/s for BJ and 1.7 ± 1.1 m/s for BD, Fig. 3). Therefore, the spatial difference of $PM_{2.5}$ and the major components between the sampling sites of BJ and the other three could not be ascribed to the difference of the wind speeds. Because the information of PBL was not available in the region of Baoding, it is difficult to discuss the impact of PBL on the spatial difference of the pollutants. According to your pertinent comment, the corresponding paragraph has been rephrased in the revised manuscript.

**Comment 4:** SOC is estimated by the EC-tracer OC/EC method. However, previous studies have demonstrated that this method could overestimate SOC under the influence of coal combustion and biomass burning, especially in wintertime. As discussed in the manuscript, coal combustion (maybe also biofuel combustion) had significant impact on PM at the sampling sites in wintertime. Thus, the concentrations and mass fractions of SOC in the winter (Figure 11) should be overestimated.

**Answer:** Yes, we totally agree with your comment. Because the lowest 10 % percentile of OC/EC ratios (3.5) measured during the sampling period were obviously less than that (13.1, Table 3) from residential coal combustion, POC could be underestimated by the product of the lowest OC/EC ratio and EC measured, and SOC could be overestimated through the subtraction of POC from OC. The overestimation of SOC by the EC-tracer OC/EC method has been noted by previous studies (Ding et al., 2012; Cui et al., 2015). The statement has been inserted in the revised manuscript.

**Comment 5:** The citation need to be re-formatted throughout the main text. For example, in line 48, the citation formats for the two references are different (Huang et al., 2014; P. Liu et al., 2016).

**Answer:** According to the citation style of the ACP, "Huang et al., 2014" is commonly the standard format. However, "P. Liu et al., 2016", "J. Liu et al., 2016" and "C. Liu et al., 2016" are often used to distinguish the references that first authors are different but both the last names of first authors and the publication years of the papers are the same.

**Comment 6:** Line 230-232. Is the difference in concentrations statistically significant? Please add p-value to show the significance of the observed difference.

**Answer:** Yes. The one-way ANOVA analysis results of the concentrations of OC, EC, $NH_4^+$, $NO_3^-$, $SO_4^{2-}$, $Cl^-$ and $K^+$ at the four sampling sites are list in Table R1. The statistically significant differences among them were found with the p-values all lower than 0.01. The p-value ($p < 0.01$) has been added in our revised manuscript.

Table R1. The one-way ANOVA analysis for the concentrations of OC, EC, $NH_4^+$, $NO_3^-$, $SO_4^{2-}$, $Cl^-$ and $K^+$ at the four sampling sites.

| | | Sum of Squares | df | Mean Square | F | Sig. |
|---|---|---|---|---|---|---|
| OC | Between Groups | 56870.407 | 3 | 18956.802 | 19.096 | .000 |
| | Within Groups | 76439.111 | 77 | 992.716 | | |

|  |  |  |  |  |  |  |
|---|---|---|---|---|---|---|
|  | Total | 133309.519 | 80 |  |  |  |
| EC | Between Groups | 3036.393 | 3 | 1012.131 | 18.014 | .000 |
|  | Within Groups | 4326.303 | 77 | 56.186 |  |  |
|  | Total | 7362.696 | 80 |  |  |  |
| $NH_4^+$ | Between Groups | 2029.908 | 3 | 676.636 | 7.820 | .000 |
|  | Within Groups | 6662.556 | 77 | 86.527 |  |  |
|  | Total | 8692.465 | 80 |  |  |  |
| $NO_3^-$ | Between Groups | 1254.055 | 3 | 418.018 | 4.188 | .008 |
|  | Within Groups | 7685.732 | 77 | 99.815 |  |  |
|  | Total | 8939.787 | 80 |  |  |  |
| $SO_4^{2-}$ | Between Groups | 2003.050 | 3 | 667.683 | 4.205 | .008 |
|  | Within Groups | 12227.563 | 77 | 158.800 |  |  |
|  | Total | 14230.613 | 80 |  |  |  |
| $Cl^-$ | Between Groups | 934.896 | 3 | 311.632 | 14.889 | .000 |
|  | Within Groups | 1611.608 | 77 | 20.930 |  |  |
|  | Total | 2546.503 | 80 |  |  |  |
| $K^+$ | Between Groups | 73.109 | 3 | 24.370 | 19.524 | .000 |
|  | Within Groups | 96.109 | 77 | 1.248 |  |  |
|  | Total | 169.218 | 80 |  |  |  |

**References**

Cui, H., Mao, P., Zhao, Y., Nielsen, C. P., and Zhang, J.: Patterns in atmospheric carbonaceous aerosols in China: emission estimates and observed concentrations, Atmospheric Chemistry and Physics, 15, 8657-8678, 10.5194/acp-15-8657-2015, 2015.

Ding, X., Wang, X. M., Gao, B., Fu, X. X., He, Q. F., Zhao, X. Y., Yu, J. Z., and Zheng, M.: Tracer-based estimation of secondary organic carbon in the Pearl River Delta, south China, Journal of Geophysical Research: Atmospheres, 117, D05313, 10.1029/2011jd016596, 2012.

Li, X., Wang, S., Duan, L., Hao, J., Li, C., Chen, Y., and Yang, L.: Particulate and Trace Gas Emissions from Open Burning of Wheat Straw and Corn Stover in China, Environ. Sci. Technol., 41, 6052-6058, 10.1021/es0705137, 2007.

Schauer, J. J., Kleeman, M. J., Cass, G. R., and Simoneit, B. R. T.: Measurement of Emissions from Air Pollution Sources. 3. $C_1-C_{29}$ Organic Compounds from Fireplace Combustion of Wood, Environ. Sci. Technol., 35, 1716-1728, 10.1021/es001331e, 2001.

Yao, L., Yang, L., Chen, J., Wang, X., Xue, L., Li, W., Sui, X., Wen, L., Chi, J., Zhu, Y., Zhang, J., Xu, C., Zhu, T., and Wang, W.: Characteristics of carbonaceous aerosols: Impact of biomass burning and secondary formation in summertime in a rural area of the North China Plain, The Science of the total environment, 557-558, 520-530, 10.1016/j.scitotenv.2016.03.111, 2016.

Zong, Z., Wang, X., Tian, C., Chen, Y., Qu, L., Ji, L., Zhi, G., Li, J., and Zhang, G.: Source apportionment of $PM_{2.5}$ at a regional background site in North China using PMF linked with radiocarbon analysis: insight into the contribution of biomass burning, Atmospheric Chemistry and Physics, 16, 11249-11265, 10.5194/acp-16-11249-2016, 2016.

---

## Author Response (AR2)

| 1        | A point-by-point response to the reviews                                                                                                                                                                                                                                                                                                                                                                                                                                                                                                                                                                                                                                                                                                                                                                                                                                                                                                                                                                                                                                                                                                                                                                                                                                                                                                                                                                                                                                                                                                                                                                                                                                                                                                                                                                                                                                                                                                                                                                                                                                                                                                                                                                                    |
|----------|-----------------------------------------------------------------------------------------------------------------------------------------------------------------------------------------------------------------------------------------------------------------------------------------------------------------------------------------------------------------------------------------------------------------------------------------------------------------------------------------------------------------------------------------------------------------------------------------------------------------------------------------------------------------------------------------------------------------------------------------------------------------------------------------------------------------------------------------------------------------------------------------------------------------------------------------------------------------------------------------------------------------------------------------------------------------------------------------------------------------------------------------------------------------------------------------------------------------------------------------------------------------------------------------------------------------------------------------------------------------------------------------------------------------------------------------------------------------------------------------------------------------------------------------------------------------------------------------------------------------------------------------------------------------------------------------------------------------------------------------------------------------------------------------------------------------------------------------------------------------------------------------------------------------------------------------------------------------------------------------------------------------------------------------------------------------------------------------------------------------------------------------------------------------------------------------------------------------------------|
| 2        | Thank you for your valuable comments. The followings are our responses to your comments.                                                                                                                                                                                                                                                                                                                                                                                                                                                                                                                                                                                                                                                                                                                                                                                                                                                                                                                                                                                                                                                                                                                                                                                                                                                                                                                                                                                                                                                                                                                                                                                                                                                                                                                                                                                                                                                                                                                                                                                                                                                                                                                                    |
| 5
4   | Response to Co-Editor                                                                                                                                                                                                                                                                                                                                                                                                                                                                                                                                                                                                                                                                                                                                                                                                                                                                                                                                                                                                                                                                                                                                                                                                                                                                                                                                                                                                                                                                                                                                                                                                                                                                                                                                                                                                                                                                                                                                                                                                                                                                                                                                                                                                       |
| 5        | Comment 1: The authors have addressed the scientific comments by the two referees in sufficient                                                                                                                                                                                                                                                                                                                                                                                                                                                                                                                                                                                                                                                                                                                                                                                                                                                                                                                                                                                                                                                                                                                                                                                                                                                                                                                                                                                                                                                                                                                                                                                                                                                                                                                                                                                                                                                                                                                                                                                                                                                                                                                      |
| 6        | detail. After reading the revised paper by myself, I have several (mostly technical) issues that still                                                                                                                                                                                                                                                                                                                                                                                                                                                                                                                                                                                                                                                                                                                                                                                                                                                                                                                                                                                                                                                                                                                                                                                                                                                                                                                                                                                                                                                                                                                                                                                                                                                                                                                                                                                                                                                                                                                                                                                                                                                                                                                      |
| 7        | should be corrected before accepting this paper for publication.                                                                                                                                                                                                                                                                                                                                                                                                                                                                                                                                                                                                                                                                                                                                                                                                                                                                                                                                                                                                                                                                                                                                                                                                                                                                                                                                                                                                                                                                                                                                                                                                                                                                                                                                                                                                                                                                                                                                                                                                                                                                                                                                                            |
| 8        |                                                                                                                                                                                                                                                                                                                                                                                                                                                                                                                                                                                                                                                                                                                                                                                                                                                                                                                                                                                                                                                                                                                                                                                                                                                                                                                                                                                                                                                                                                                                                                                                                                                                                                                                                                                                                                                                                                                                                                                                                                                                                                                                                                                                                             |
| 9        | Answer: Thank you very much for your appreciation. The followings are our responses to your                                                                                                                                                                                                                                                                                                                                                                                                                                                                                                                                                                                                                                                                                                                                                                                                                                                                                                                                                                                                                                                                                                                                                                                                                                                                                                                                                                                                                                                                                                                                                                                                                                                                                                                                                                                                                                                                                                                                                                                                                                                                                                                                 |
| 10       | comments.                                                                                                                                                                                                                                                                                                                                                                                                                                                                                                                                                                                                                                                                                                                                                                                                                                                                                                                                                                                                                                                                                                                                                                                                                                                                                                                                                                                                                                                                                                                                                                                                                                                                                                                                                                                                                                                                                                                                                                                                                                                                                                                                                                                                                   |
| 11       |                                                                                                                                                                                                                                                                                                                                                                                                                                                                                                                                                                                                                                                                                                                                                                                                                                                                                                                                                                                                                                                                                                                                                                                                                                                                                                                                                                                                                                                                                                                                                                                                                                                                                                                                                                                                                                                                                                                                                                                                                                                                                                                                                                                                                             |
| 12       | Comment 2: First, in section 3.4 and elsewhere, the authors should not confuse between the strength                                                                                                                                                                                                                                                                                                                                                                                                                                                                                                                                                                                                                                                                                                                                                                                                                                                                                                                                                                                                                                                                                                                                                                                                                                                                                                                                                                                                                                                                                                                                                                                                                                                                                                                                                                                                                                                                                                                                                                                                                                                                                                                  |
| 13       | (R values) and significance (p values) of the correlations when discussing the results. Now,                                                                                                                                                                                                                                                                                                                                                                                                                                                                                                                                                                                                                                                                                                                                                                                                                                                                                                                                                                                                                                                                                                                                                                                                                                                                                                                                                                                                                                                                                                                                                                                                                                                                                                                                                                                                                                                                                                                                                                                                                                                                                                                                |
| 14       | throughout section 3.4, the authors refer to significance of the correlation even though in most cases                                                                                                                                                                                                                                                                                                                                                                                                                                                                                                                                                                                                                                                                                                                                                                                                                                                                                                                                                                                                                                                                                                                                                                                                                                                                                                                                                                                                                                                                                                                                                                                                                                                                                                                                                                                                                                                                                                                                                                                                                                                                                                                      |
| 15       | it is actually the strength of the correlation that is meant (e.g. in figure 10 the correlation is moderate                                                                                                                                                                                                                                                                                                                                                                                                                                                                                                                                                                                                                                                                                                                                                                                                                                                                                                                                                                                                                                                                                                                                                                                                                                                                                                                                                                                                                                                                                                                                                                                                                                                                                                                                                                                                                                                                                                                                                                                                                                                                                                                 |
| 16       | to strong for 6 cases and non-existent for the 2 other cases). Please correct.                                                                                                                                                                                                                                                                                                                                                                                                                                                                                                                                                                                                                                                                                                                                                                                                                                                                                                                                                                                                                                                                                                                                                                                                                                                                                                                                                                                                                                                                                                                                                                                                                                                                                                                                                                                                                                                                                                                                                                                                                                                                                                                                              |
| 17
18 | Answer: Thank you for your valuable guidance. The strength and significance of the correlations                                                                                                                                                                                                                                                                                                                                                                                                                                                                                                                                                                                                                                                                                                                                                                                                                                                                                                                                                                                                                                                                                                                                                                                                                                                                                                                                                                                                                                                                                                                                                                                                                                                                                                                                                                                                                                                                                                                                                                                                                                                                                                                             |
| 19       | have been distinguished in our revised manuscript. When discussing the R values "the significant                                                                                                                                                                                                                                                                                                                                                                                                                                                                                                                                                                                                                                                                                                                                                                                                                                                                                                                                                                                                                                                                                                                                                                                                                                                                                                                                                                                                                                                                                                                                                                                                                                                                                                                                                                                                                                                                                                                                                                                                                                                                                                                            |
| 20       | correlations" has been revised as "the strong correlations": when discussing the p values, the p                                                                                                                                                                                                                                                                                                                                                                                                                                                                                                                                                                                                                                                                                                                                                                                                                                                                                                                                                                                                                                                                                                                                                                                                                                                                                                                                                                                                                                                                                                                                                                                                                                                                                                                                                                                                                                                                                                                                                                                                                                                                                                                            |
| 21       | values has been inserted in the manuscript.                                                                                                                                                                                                                                                                                                                                                                                                                                                                                                                                                                                                                                                                                                                                                                                                                                                                                                                                                                                                                                                                                                                                                                                                                                                                                                                                                                                                                                                                                                                                                                                                                                                                                                                                                                                                                                                                                                                                                                                                                                                                                                                                                                                 |
| 22       | 1                                                                                                                                                                                                                                                                                                                                                                                                                                                                                                                                                                                                                                                                                                                                                                                                                                                                                                                                                                                                                                                                                                                                                                                                                                                                                                                                                                                                                                                                                                                                                                                                                                                                                                                                                                                                                                                                                                                                                                                                                                                                                                                                                                                                                           |
| 23       | Comment 3: Second, the comparison of percentages on lines 541-545 is not quite correct. E.g. if                                                                                                                                                                                                                                                                                                                                                                                                                                                                                                                                                                                                                                                                                                                                                                                                                                                                                                                                                                                                                                                                                                                                                                                                                                                                                                                                                                                                                                                                                                                                                                                                                                                                                                                                                                                                                                                                                                                                                                                                                                                                                                                             |
| 24       | the proportion in A is 12% and in B it is 10%, then this proportion is not 2% higher in A compared                                                                                                                                                                                                                                                                                                                                                                                                                                                                                                                                                                                                                                                                                                                                                                                                                                                                                                                                                                                                                                                                                                                                                                                                                                                                                                                                                                                                                                                                                                                                                                                                                                                                                                                                                                                                                                                                                                                                                                                                                                                                                                                          |
| 25       | with B. Please give only absolute percentages of these proportions, and when comparing different                                                                                                                                                                                                                                                                                                                                                                                                                                                                                                                                                                                                                                                                                                                                                                                                                                                                                                                                                                                                                                                                                                                                                                                                                                                                                                                                                                                                                                                                                                                                                                                                                                                                                                                                                                                                                                                                                                                                                                                                                                                                                                                            |
| 26       | sites, simple state qualitatively to which direction they are between different sites (e.g. slightly larger                                                                                                                                                                                                                                                                                                                                                                                                                                                                                                                                                                                                                                                                                                                                                                                                                                                                                                                                                                                                                                                                                                                                                                                                                                                                                                                                                                                                                                                                                                                                                                                                                                                                                                                                                                                                                                                                                                                                                                                                                                                                                                                 |
| 27       | than).                                                                                                                                                                                                                                                                                                                                                                                                                                                                                                                                                                                                                                                                                                                                                                                                                                                                                                                                                                                                                                                                                                                                                                                                                                                                                                                                                                                                                                                                                                                                                                                                                                                                                                                                                                                                                                                                                                                                                                                                                                                                                                                                                                                                                      |
| 28       |                                                                                                                                                                                                                                                                                                                                                                                                                                                                                                                                                                                                                                                                                                                                                                                                                                                                                                                                                                                                                                                                                                                                                                                                                                                                                                                                                                                                                                                                                                                                                                                                                                                                                                                                                                                                                                                                                                                                                                                                                                                                                                                                                                                                                             |
| 29       | Answer: Thank you for your valuable comment. The sentence has been rephrased as "The average                                                                                                                                                                                                                                                                                                                                                                                                                                                                                                                                                                                                                                                                                                                                                                                                                                                                                                                                                                                                                                                                                                                                                                                                                                                                                                                                                                                                                                                                                                                                                                                                                                                                                                                                                                                                                                                                                                                                                                                                                                                                                                                                |
| 30       | mass proportions of OC and EC at BD, WD and DBT were very close, accounting for about 45.7 %-                                                                                                                                                                                                                                                                                                                                                                                                                                                                                                                                                                                                                                                                                                                                                                                                                                                                                                                                                                                                                                                                                                                                                                                                                                                                                                                                                                                                                                                                                                                                                                                                                                                                                                                                                                                                                                                                                                                                                                                                                                                                                                                               |
| 31       | 47.1 % and 9.0 %-10.4 % of the total species in PM 2.5 , respectively, which were much greater than $(27.0.0)$ for OC and $7.4.0$ (for PC) of PL L and $(10.0)$ ( $10.0$ m) |
| 32
22 | those (37.9% for UC and 7.4% for EC) at BJ. In contrast to UC and EC, the average mass properties of NO- $(10.1\% + 10.8\%)$ and SO $(2-(11.2\% + 11.7\%))$ at PD. WD and DPT were slightly                                                                                                                                                                                                                                                                                                                                                                                                                                                                                                                                                                                                                                                                                                                                                                                                                                                                                                                                                                                                                                                                                                                                                                                                                                                                                                                                                                                                                                                                                                                                                                                                                                                                                                                                                                                                                                                                                                                                                                                                                                 |
| 21       | proportions of NO 3 (10.1 %-10.8 %) and SO 4 - (11.2 %-11.7 %) at BD, wD and DB1 were signify
less than those (15.1 % for NO 2 and 14.0 % for SO 4 2- ) at BL respectively." in our revised manuscript                                                                                                                                                                                                                                                                                                                                                                                                                                                                                                                                                                                                                                                                                                                                                                                                                                                                                                                                                                                                                                                                                                                                                                                                                                                                                                                                                                                                                                                                                                                                                                                                                                                                                                                                                                                                                                                                                                                                                 |
| 34
35 | less than those (15.1 % for two3 and 14.0 % for 504 ) at b3, respectively. In our revised manuscript.                                                                                                                                                                                                                                                                                                                                                                                                                                                                                                                                                                                                                                                                                                                                                                                                                                                                                                                                                                                                                                                                                                                                                                                                                                                                                                                                                                                                                                                                                                                                                                                                                                                                                                                                                                                                                                                                                                                                                                                                                                                                                                                       |
| 36       | Comment 4: Third, there are a number of sentences that are unclear and should be modified, either                                                                                                                                                                                                                                                                                                                                                                                                                                                                                                                                                                                                                                                                                                                                                                                                                                                                                                                                                                                                                                                                                                                                                                                                                                                                                                                                                                                                                                                                                                                                                                                                                                                                                                                                                                                                                                                                                                                                                                                                                                                                                                                    |
| 37       | grammatically or otherwise. I list their locations below:                                                                                                                                                                                                                                                                                                                                                                                                                                                                                                                                                                                                                                                                                                                                                                                                                                                                                                                                                                                                                                                                                                                                                                                                                                                                                                                                                                                                                                                                                                                                                                                                                                                                                                                                                                                                                                                                                                                                                                                                                                                                                                                                                                   |
| 38       | 320-323                                                                                                                                                                                                                                                                                                                                                                                                                                                                                                                                                                                                                                                                                                                                                                                                                                                                                                                                                                                                                                                                                                                                                                                                                                                                                                                                                                                                                                                                                                                                                                                                                                                                                                                                                                                                                                                                                                                                                                                                                                                                                                                                                                                                                     |
| 39       | Answer: "the promotion new stoves" has been revised as "the promoted new stoves".                                                                                                                                                                                                                                                                                                                                                                                                                                                                                                                                                                                                                                                                                                                                                                                                                                                                                                                                                                                                                                                                                                                                                                                                                                                                                                                                                                                                                                                                                                                                                                                                                                                                                                                                                                                                                                                                                                                                                                                                                                                                                                                                           |
| 40       | 480-481 (what is meant by fluctuating trends?)                                                                                                                                                                                                                                                                                                                                                                                                                                                                                                                                                                                                                                                                                                                                                                                                                                                                                                                                                                                                                                                                                                                                                                                                                                                                                                                                                                                                                                                                                                                                                                                                                                                                                                                                                                                                                                                                                                                                                                                                                                                                                                                                                                              |
| 41       | Answer: "exhibited similar fluctuation trends" has been revised as "exhibited similar trend".                                                                                                                                                                                                                                                                                                                                                                                                                                                                                                                                                                                                                                                                                                                                                                                                                                                                                                                                                                                                                                                                                                                                                                                                                                                                                                                                                                                                                                                                                                                                                                                                                                                                                                                                                                                                                                                                                                                                                                                                                                                                                                                               |
| 42       | 487 (are you referring to pollutant concentrations here, please specify)                                                                                                                                                                                                                                                                                                                                                                                                                                                                                                                                                                                                                                                                                                                                                                                                                                                                                                                                                                                                                                                                                                                                                                                                                                                                                                                                                                                                                                                                                                                                                                                                                                                                                                                                                                                                                                                                                                                                                                                                                                                                                                                                                    |
| 43       | Answer: "spatial and temporal difference of pollutants" has been revised as "spatial and temporal                                                                                                                                                                                                                                                                                                                                                                                                                                                                                                                                                                                                                                                                                                                                                                                                                                                                                                                                                                                                                                                                                                                                                                                                                                                                                                                                                                                                                                                                                                                                                                                                                                                                                                                                                                                                                                                                                                                                                                                                                                                                                                                           |

- 44 difference in concentrations of pollutants".
- 45 505-507
- 46 Answer: The sentence has been rephrased as "Compared with the cities, the distinct source for
- 47 atmospheric pollutants at DBT in winter is the residential coal combustion because residential coal
- 48 **combustion** is prevailingly used for heating and cooking in rural areas of the Northern China."
- 49 528-530 (high density of countryside?)
- Answer: "high density of countryside" has been revised as "countryside with high farmer density".
  51 569-571
- 52 Answer: The sentence has been rephrased as "Because the average concentrations of the species
- 53 in PM2.5 were mainly controlled by the highest concentration values and the relatively high
- 54 concentration level of the species in PM2.5 at BJ usually occurred during the serious pollution
- episodes, the proportions of the species in PM2.5 were dominated by the serious pollution events."
  588-590
- Answer: The sentence has been rephrased as "It should be mentioned that the OC/EC ratios observed at DBT and WD were about a factor of 2.7 less than that (13.1) of the emission from the residential coal combustion and, however, the OC/EC ratios observed at BJ and BD were too high to be explained by direct emissions from diesel (0.4-0.8) and gasoline (3.1) vehicles (Shah et
- 61 al., 2004; Geller et al., 2006)."
- 62 685-687
- Answer: The sentence has been rephrased as "If the primary PM2.5 was only considered, the contribution of residential coal combustion to the primary PM2.5 at BJ would achieve to be about
  59 %, which was in line with the value of 57 % estimated by J. Liu et al. (2016) for the winter of 2010 in Beijing."
- 67
- 68 **Comment 5:** Finally, there are a number of minor technical and grammatical issues that should be 69 corrected:
- 70 Throughout the paper: difference in, not of
- 71 Line 271: ...sites were almost the same (4.8) when ....
- 72 277-278: ...rural areas, whereas...transportation, were...
- 73 290-291: ... matter with an aerodynamic diameter...
- 74 303: ...levels can still be larger than 1000...
- 75 309: ... combustion, which ... region, was...
- 76 324: There are...
- 338-340: please delete the last 4 "the" from the list (only the first one should be there). Also delete
- 78 "method" from the end.
- 79 364: delete "p.m." 15:00 already reveals that it is in the afternoon
- 80 367: 10 mL of ultrapure
- 81 369: ... before the analysis, and the ...
- 82 374: ... 10%, and the ....
- 83 377: in the same way
- 84 383: A chemical mass closure...
- 438-440: Meteorological data, including...temperature and barometric pressure, as well as air
- quality index (AQI) based on PM ... and WD, were obtained....
- 87 450: ... for each sampling day.

- 465: delete "obviously". The values are, or are not, positive.
- 89 481: differences
- 90 483: especially the wind speed
- 91 488-489: ...considered similar because...
- 92 508-509: ...due to the lack of any control measures, as strong...
- 93 513-514: please move the citation to the end of this sentence
- 94 515-516: ... process can be as high as... (Is this what you mean here?)
- 95 575: should this be "relations" rather than "correlations"?
- 96 584-585: ...a dominant...periods.
- 97 594: ... very reactive, favoring secondary organic aerosol (SOA) formation (Zhang...)
- 98 600: ...would be smaller during...
- 99 630: have been reported to be...
- 100 632: which is at least...
- 101
- Answer: Thank you very much for your careful reviews. These mistakes have been corrected in our revised manuscript:
- 104 Throughout the paper: "difference of" has been revised as "difference in"
- Line 271: "...sites became the almost same value of 4.8..." has been revised as "...sites were almost
  the same (4.8) when ..."
- 277-278: The sentence has been rephrased as "...residential coal combustion was the dominant
  source for the key species in the rural area and, however, the complex sources including local
- 109 emissions and regional transportation were **responsible** for **the** atmospheric species in the cities"
- 290-291: "...fine particulate matters with dynamic diameter..." has been revised as "...matter withan aerodynamic diameter..."
- 112 303: "...levels still achieved to be above..." has been revised as "...levels can still be larger than..."
- 309: "...combustion which ... region was..." has been revised as "...combustion, which ... region,
  was..."
- 115 324: "There were..." has been revised as "There are..."
- 116 338-340: The sentence has been rephrased as "...based on the  $PM_{2.5}$  levels,  $PM_{2.5}$  composition 117 characteristics, correlations among key species in  $PM_{2.5}$ , back trajectories and chemical mass 118 closure."
- 119 364: "p.m." has been deleted in our revised manuscript
- 120 367: "...10 mL ultrapure..." has been revised as "...10 mL of ultrapure..."
- 121 369: "... before analysis and the ..." has been revised as "... before the analysis, and the ..."
- 122 374: "... 10% and the ...." has been revised as "... 10%, and the ...."
- 123 377: "as the same way" has been revised as "in the same way"
- 124 383: "Chemical mass closure..." has been revised as "A chemical mass closure..."
- 125 438-440: Both the meteorological data, including ... temperature, barometric pressure and air quality
- 126 index (AQI) of PM ... and WD were obtained...." has been revised as "Meteorological data,
- 127 including...temperature and barometric pressure, as well as air quality index (AQI) based on PM ...
- 128 and WD, were obtained...."
- 129 450: "respectively" has been deleted in our revised manuscript
- 130 465: "obviously" has been deleted in our revised manuscript
- 481: "...there was obvious difference..." has been revised as "...there were obvious differences..."

| 132 | 483: "especially win | d speed" has be | en revised as | "especially the win | d speed" |
|-----|----------------------|-----------------|---------------|---------------------|----------|
|-----|----------------------|-----------------|---------------|---------------------|----------|

488-489: "...considered as the same because..." has been revised as "...considered similarbecause..."

- 135 508-509: "...due to lack of any control measures, strong..." has been revised as "...due to the lack136 of any control measures, as strong..."
- 137 513-514: the citation has been moved to the end of this sentence in our revised manuscript
- 138 515-516: "...process could achieve to be" has been revised as "... process can be as high as..."

139 575: "correlations" has been corrected as "relations"

140 584-585: "...made dominant contribution..." has been revised as "...made a dominant
141 contribution..."

- 142 594: "...very reactive to make contribution to secondary organic aerosols (SOA) (Zhang...)" has
- been revised as "... very reactive, favoring secondary organic aerosol (SOA) formation (Zhang...)"
- 144 600: "...would become less during..." has been revised as "...would be smaller during..."
- 145 630: "were reported to be..." has been revised as "have been reported to be..."
- 146 632: "...which were at least..." has been revised as "...which is at least..."
- 147 Thank you very much for all you've done for us.

| 176 | A list of all relevant changes made in the manuscript                                              |
|-----|----------------------------------------------------------------------------------------------------|
| 177 | Based on the valuable comments and suggestions of the Co-editor, the followings are a list of all  |
| 178 | relevant changes made in the manuscript.                                                           |
| 179 |                                                                                                    |
| 180 | 1. The strength and significance of the correlations have been distinguished and corrected in our  |
| 181 | revised manuscript.                                                                                |
| 182 | 2. The discussion about the comparison of percentages has been improved in our revised manuscript. |
| 183 | 3. A number of sentences that were unclear have been modified and rephrased in our revised         |
| 184 | manuscript.                                                                                        |
| 185 | 4. Many logical and grammatical mistakes have been corrected in our revised manuscript.            |
| 186 |                                                                                                    |
| 187 |                                                                                                    |
| 188 |                                                                                                    |
| 189 |                                                                                                    |
| 190 |                                                                                                    |
| 191 |                                                                                                    |
| 192 |                                                                                                    |
| 193 |                                                                                                    |
| 194 |                                                                                                    |
| 195 |                                                                                                    |
| 196 |                                                                                                    |
| 197 |                                                                                                    |
| 198 |                                                                                                    |
| 199 |                                                                                                    |
| 200 |                                                                                                    |
| 201 |                                                                                                    |
| 202 |                                                                                                    |
| 203 |                                                                                                    |
| 204 |                                                                                                    |
| 205 |                                                                                                    |
| 206 |                                                                                                    |
| 207 |                                                                                                    |
| 208 |                                                                                                    |
| 209 |                                                                                                    |
| 210 |                                                                                                    |
| 211 |                                                                                                    |
| 212 |                                                                                                    |
| 213 |                                                                                                    |
| 215 |                                                                                                    |
| 215 |                                                                                                    |
| 217 |                                                                                                    |
| 218 |                                                                                                    |
| 219 |                                                                                                    |
| -   |                                                                                                    |

**The contribution of residential coal combustion to atmospheric $PM_{2.5}$**

**in the North China during winter**

**Pengfei Liu1, 3, Chenglong Zhang1, 2, 3, Chaoyang Xue1, 3, Yujing Mu1, 2, 3, 4, Junfeng Liu1, 2, 3, Yuanyuan Zhang1, 2, 3, Di Tian1, 3, Can Ye1, 3, Hongxing Zhang1, 5, Jian Guan6**

[revised manuscript text omitted]

 (2)

374
$$\frac{\left[Cl_{cc}^{-}\right]_{35.5}}{\left[Na_{cc}^{+}\right]_{23}} = 1.4$$
(3)

375
$$\frac{\left[Cl_{ss}^{-}\right]}{\left[Na_{ss}^{+}\right]}_{23} = 1.18$$
(4)

376 where [Cl-ss] and [Na+ss] are the mass concentrations of Cl- and Na+ from sea salt, and [Cl-cc] and [Na+cc] are the mass concentrations of Cl- and Na+ from coal combustion. The molar ratio of Cl-ss to 377 Na+ss was adopted to be 1.18 which represented the typical ratio from sea salt (Brewer, 1975). The 378 379 molar ratio of Cl-cc to Na+cc was chosen to be 1.4 in this study according to our preliminary 380 measurements from the raw bituminous coal prevailed in the North China and the value of 1.4 has 381 been recorded by the previous study (Bl äsing and Müller, 2012). If the molar ratios of atmospheric 382 Cl- to Na+ in PM2.5 were greater than the value of 1.4 or lower than the value of 1.18, atmospheric 383 Cl- and Na+ would be considered to be totally from coal combustion or sea salt.

Because the average Al content accounts for about 7 % in mineral dust (Zhang et al., 2003; Ho et al., 2006; Hsu et al., 2010a; Zhang et al., 2013), the mineral dust was estimated based on the follow

386 equation:

$$[Mineral \,dust] = \frac{[Al]}{0.07} \tag{5}$$

POC and SOC were calculated by the EC-tracer OC/EC method (Cheng et al., 2011; Zhao et al.,
2013b; G. J. Zheng et al., 2015; Cui et al., 2015) as follows:

390
$$[POC] = [EC] \times ({[OC]/_{[EC]}})_{pri} = K[EC] + M$$
 (6)

$$\quad [SOC] = [OC] - [POC] \tag{7}$$

392 The values of K and M are estimated by linear regression analysis using the data pairs with the

lowest 10 % percentile of ambient OC/EC ratios. It should be mentioned that POC could be underestimated and SOC could be overestimated by the EC-tracer OC/EC method, because the lowest 10 % percentile of OC/EC ratios measured were usually less than those from dominant sources of coal combustion and biomass burning in autumn and winter (Ding et al., 2012; Cui et al., 2015).

To estimate the contribution of heavy metal oxide, the enrichment factors (EF) of various heavy metal elements were calculated by the following equation (Hsu et al., 2010b; Zhang et al., 2013):

400
$$EF = \frac{\left(\frac{[Element]}{[Al]}\right)_{aerosol}}{\left(\frac{[Element]}{[Al]}\right)_{crust}}$$
(8)

[revised manuscript text omitted]

- 1017
- 1018
- 1019
- 1020
- 1021
- 1022